# RELATIVE ENTROPY PATHWISE POLICY OPTIMIZATION

**Claas A Voelcker**[1,2,*]**, Axel Brunnbauer**[3,*]**, Marcel Hussing**[4]**, Michal Nauman**[5,6]**,
Pieter Abbeel**[6]**, Eric Eaton**[4]**, Radu Grosu**[3]**, Amir-massoud Farahmand**[7,8,1]**,
Igor Gilitschenski**[1,2]
[1]University of Toronto, [2]Vector Institute, [3]TU Wien, [4]University of Pennsylvania
[5]University of Warsaw, [6]UC Berkeley, [7]Polytechnique Montréal, [8]Mila – Quebec AI Institute
[*] Authors contributed equally, correspondence to `cvoelcker@cs.toronto.edu`,
`axel.brunnbauer@tuwien.ac.at`

## ABSTRACT

Score-function based methods for policy learning, such as REINFORCE and PPO, have delivered strong results in game-playing and robotics, yet their high variance often undermines training stability. Improving a policy through state-action value functions, for example by differentiating Q with regard to the policy, alleviates the variance issues. However, this requires an accurate action-conditioned value function, which is notoriously hard to learn without relying on replay buffers for reusing past off-policy data. We present Relative Entropy Pathwise Policy Optimization, an algorithm that trains Q-value models purely from on-policy trajectories, unlocking the use of Q function derivatives to compute policy updates in the context of on-policy learning. We show how to combine stochastic policies for exploration with constrained updates for stable training, and evaluate important architectural components that stabilize value function learning. This results in an efficient on-policy algorithm that combines the stability of Q-based policy gradients with the simplicity and minimal memory footprint of standard on-policy learning. Compared to state-of-the-art on two standard GPU-parallelized benchmarks, REPPO provides strong empirical performance at superior sample efficiency, wall-clock time, memory footprint, and hyperparameter robustness.

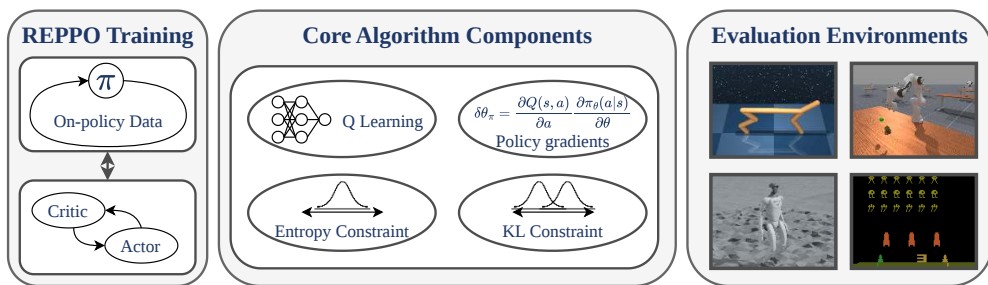

Figure 1: Overview of the **Relative Entropy Pathwise Policy Optimization** algorithm. REPPO combines four core components to achieve stable and fast actor-critic training across a wide variety of reinforcement learning tasks. Isaac and Atari results can be found in the appendix. Implementations in different popular RL frameworks can be found at `https://github.com/reppo-rl`.

## 1 INTRODUCTION

Most modern on-policy algorithms, such as TRPO (Schulman et al., 2015) or PPO (Schulman et al., 2017), use a score-based gradient estimator to update the policy. These methods have proven useful for robotic control (Rudin et al., 2022; Kaufmann et al., 2023; Radosavovic et al., 2024), and language-model fine-tuning (Ouyang et al., 2022; Touvron et al., 2023; Gao et al., 2023; Liu et al.,

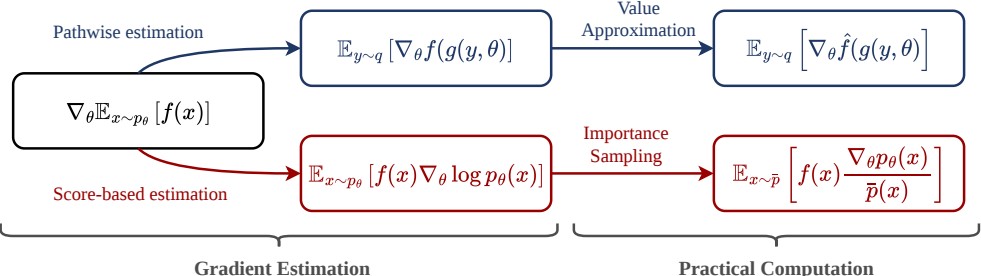

Figure 2: Overview of the strategies used by REPPO and PPO to obtain policy gradient estimators. Computing the gradient requires a mathematical transformation that allows for efficient estimation from samples, and additional steps that make the computation tractable in practice.

2024), but are often plagued by training instability. Zeroth-order, score-based gradient approximation exhibits high variance (Greensmith et al., 2004), which leads to unstable learning (Ilyas et al., 2020; Rahn et al., 2023), especially in high-dimensional continuous spaces (Li et al., 2018). In addition, it requires importance sampling to allow sample reuse, which exacerbates the high variance.

Alternatively we can train a parameterized state-action value function (Lillicrap et al., 2016; Fujimoto et al., 2018; Haarnoja et al., 2018), and use it to improve the policy, for example by using a *pathwise* policy gradient (Silver et al., 2014), i.e. taking the derivative of the Q function wrt the action. Using a parameterized function to improve the policy often leads to faster and more stable learning learning by reducing the score-based estimators variance (Mohamed et al., 2020) and by allowing us to remove importance sampling corrections.

However, the effectiveness of these approaches is bounded by the quality of the approximate value function (Silver et al., 2014). As such, algorithms that use a state-action value function usually rely on improving value learning through off-policy training (Fujimoto et al., 2018; Haarnoja et al., 2018). Unfortunately, off-policy training requires the use of replay buffers. Storing these replay buffers can be a challenge when the collected samples cannot fit in memory. In addition, training with past data introduces various challenges for value function fitting (Thrun & Schwartz, 1993; Baird, 1995; Van Hasselt, 2010; Sutton et al., 2016; Kumar et al., 2021; Nikishin et al., 2022; Lyle et al., 2024; Hussing et al., 2024; Voelcker et al., 2025). This raises our core question:

> *Can we train a sufficiently correct value function approximation and effectively use it for policy improvement in a fully on-policy setting without large replay buffers?*

Building on the progress in accurate value function learning (Sutton, 1988; Haarnoja et al., 2019; Schwarzer et al., 2021; Hussing et al., 2024; Farebrother et al., 2024), we present an efficient on-policy algorithm, *Relative Entropy Pathwise Policy Optimization (REPPO)*, which uses the pathwise gradient estimator with an accurate value estimation trained on on-policy data. REPPO builds on the maximum entropy framework (Ziebart et al., 2008) to encourage exploration. It combines this with a KL regularization scheme, inspired by the Relative Entropy Policy Search method (Peters et al., 2010), which prevents aggressive policy updates from destabilizing the optimization.

Furthermore, we evaluate several prominent advances in neural network architecture design to stabilize learning: categorical Q-learning (Farebrother et al., 2024), normalized neural network architectures (Nauman et al., 2024a; Hussing et al., 2024), and auxiliary tasks (Jaderberg et al., 2017). These components feature in many recent variants (Schwarzer et al., 2021; 2023; Nauman et al., 2024a; Hussing et al., 2024; Gallici et al., 2024; Lee et al., 2025a;b; Nauman et al., 2025; Fujimoto et al., 2024) of common value learning algorithm such as SAC (Haarnoja et al., 2018). We find that categorical Q-learning and normalization have a strong impact on the performance, while auxiliary tasks only show small impact, but become more relevant when reducing the amount of samples.

We test our approach in a variety of locomotion and manipulation environments from the Mujoco Playground (Zakka et al., 2025) and ManiSkill (Tao et al., 2025) benchmarks, and show that REPPO is competitive with tuned on-policy baselines in terms of sample efficiency and wall-clock time, while using significantly smaller memory footprints than comparable off-policy algorithms. Fur-

thermore, we find that the proposed method is robust to the choice of hyperparameters. To this end, our method offers stable performance across more than 30 tasks spanning multiple benchmarks with a single hyperparameter set. In introducing REPPO, our work makes the following contributions:

1. We showcase that using a state-action value function and a pathwise policy gradient can be effective in on-policy RL, as it allows on-policy action resampling, forgoing importance corrections. However, this requires learning a highly accurate state-action value function.

2. We show how a joint entropy and policy deviation tuning objective can address the twin problems of sufficient exploration and controlled policy updates.

3. We evaluate architectural components such as cross-entropy losses, layer normalization, and auxiliary tasks for their efficacy in pathwise policy gradient-based on-policy learning.

We provide sample implementations in both the JAX (Bradbury et al., 2018) and PyTorch (Paszke et al., 2019) frameworks at `https://github.com/reppo-rl`.

## 2 BACKGROUND, NOTATION, AND DEFINITIONS

We consider the setting of the Markov Decision Process (MDP) (Puterman, 1994) , defined by the tuple $(\mathcal{X}, \mathcal{A}, \mathcal{P}, r, \gamma, \rho_0)$, where $\mathcal{X}$ is the set of states, $\mathcal{A}$ is the set of actions, $\mathcal{P}(x'|x, a)$ is the transition probability kernel, $r(x, a)$ is the reward function, and $\gamma \in [0, 1)$ is the discount factor. We write $\mathcal{P}_\pi(x'|x)$ for the policy-conditioned transition kernel and $\mathcal{P}_\pi^n(y|x)$ for the n-step transition kernel. An agent interacts with the environment via a policy $\pi(a|x)$, which defines a distribution over actions given a state. The objective is to find a policy that maximizes the expected discounted return, $J(\pi) = \mathbb{E}_\pi \left[ \sum_{t=0}^\infty \gamma^t r(x_t, a_t) \right]$, where $x_0 \sim \rho_0$ is the initial state distribution, and $a_t \sim \pi(\cdot|x_t)$. The state-action value function associated with a policy $\pi$ are defined as $Q^\pi(x, a) = \mathbb{E}_\pi \left[ \sum_{t=0}^\infty \gamma^t r(x_t, a_t) \Big| x_0 = x, a_0 = a \right]$. We use $\mu_\pi(y|x)$ to denote the discounted stationary distribution over states $y$ when starting in state $x$. When $x \sim \mu_\pi(\cdot|y), y \sim \rho_0$, we will simply write $\mu_\pi(x)$ to denote the probability of a state under the discounted occupancy distribution.[1]

### 2.1 POLICY GRADIENT LEARNING

A policy gradient approach (Sutton & Barto, 2018) is a general method for improving a (parameterized) policy $\pi_\theta$ by estimating the gradient of the policy-return function $J(\pi_\theta)$ with regard to the policy parameters $\theta$. The *policy gradient theorem* states that

$$\nabla_\theta J(\pi_\theta) = \mathbb{E}_{x \sim \mu_\pi, a \sim \pi_\theta(\cdot|x)}[Q^{\pi_\theta}(x, a)\nabla_\theta \log \pi_\theta(a|x)]. \tag{1}$$

This identity is particularly useful as both the Q value and the stationary distribution can be estimated by samples obtained from following the policy for sufficiently many steps in the environment.

An alternative approach is the *deterministic policy gradient theorem* (DPG) (Silver et al., 2014). The estimator for the DPG relies on access to a differentiable state-action value function and a deterministic differentiable policy $\pi_\theta^{\text{det}}(x)$. While access to the true value function is an unrealistic assumption, we can use a trained surrogate model, $\hat{Q}$, to obtain a biased estimate of the gradient

$$\nabla_\theta J(\pi_\theta) \approx \mathbb{E}_{x \sim \mu_\pi}[\nabla_a \hat{Q}^{\pi_\theta^{\text{det}}}(x, a)|_{a=\pi_\theta^{\text{det}}(x)} \nabla_\theta \pi_\theta^{\text{det}}(x)]. \tag{2}$$

Finally, the DPG can be expanded to reparametrizable stochastic policies[2]. We term this the *pathwise policy gradient*, following Mohamed et al. (2020), but the formulation has been used prominently in prior work such as SAC (Haarnoja et al., 2018), just without a proper name. The gradient estimator can be obtained from the following expectation

$$\nabla_\theta J(\pi_\theta) \approx \mathbb{E}_{x \sim \mu_\pi, \epsilon \sim p(\epsilon)}[\nabla_a \hat{Q}^{\pi_\theta^{\text{rep}}}(x, a)|_{a=\pi_\theta^{\text{rep}}(x, \epsilon)} \nabla_\theta \pi_\theta^{\text{rep}}(x, \epsilon)], \tag{3}$$

where $\pi_\theta^{\text{rep}}(x, \epsilon)$ is a reparameterization of $\pi_\theta(a|x)$. To avoid notational clutter we will write $\pi_\theta(a|x)$ from now on to always mean the appropriate reparameterization.

---

[1] Many policy gradient methods use the undiscounted empirical state occupancy for optimization (Nota & Thomas, 2020). While this could be considered an error, it is nonetheless a common approximation. REPPO similarly uses empirical samples without accounting for the discount factor in the objective.

[2] We discuss an extension to non-reparametrizable, discrete policies in Appendix C.

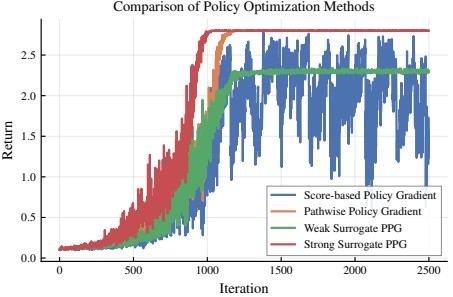
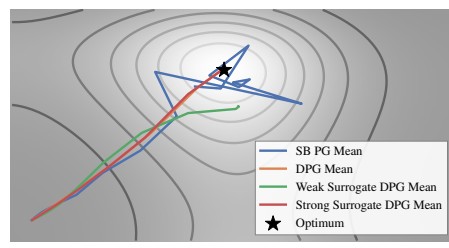

(a) Achieved returns (left) and path of four policies trained with different gradient estimation methods. We compare a score-function based policy gradient estimator (blue) with three variants of pathwise gradient estimators: using the ground truth objective function (orange), an inaccurate surrogate model (green), and an accurate surrogate model (red). All PPG based methods show markedly reduced variance in the policy updates.

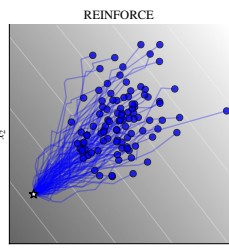
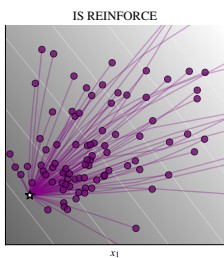
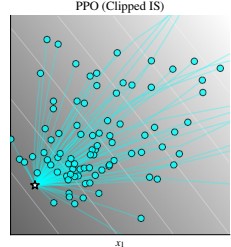
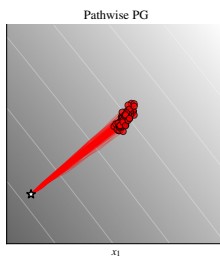

(b) Gradient path over eight steps in the middle of the trajectory, visualized per algorithm for 8 steps. For Reinforce and PPG, new samples are drawn at every step. For the importance sampling based algorithms, one set of samples is sampled at the beginning and subsequent steps are conducted using importance sampling.

Figure 3: Visualization of gradient paths on a 2D example function.

## 2.2 Understanding sources of harmful variance in gradient estimation

To build additional intuition on the differences between different policy gradient estimators, we conduct an illustrative experiment. Implementation details can be found in Appendix D.

We initialize four Gaussians and update their parameters to maximize $J(\mu, \Sigma) = \mathbb{E}_{x \sim \mathcal{N}(\cdot|\mu,\Sigma)}[g(x)]$ on a test function $g(x)$ with four different methods: a score-based policy gradient (using Equation 1), a pathwise PG with the ground truth objective function, and two pathwise PGs using learned approximations, one accurate and one inaccurate (all using Equation 3). We visualize the returns and the path of the mean estimates in Figure 3a. In addition, we zoom in on the gradient paths of the score-based estimator. We visualize 100 different eight step paths from the middle of the trajectory. Here, in addition to the vanilla score-based estimator, we also show an importance sampling and a clipped importance sampling estimator. These paths are visualized in Figure 3b.

The experiment shows that score-based gradient estimators have high variance, and can lead to unstable policies which fail to optimize the target. In addition, while importance sampling increases the sample efficiency of the algorithm, it greatly exacerbates these variance issues. We find that clipping the ratio estimate, as proposed by Schulman et al. (2017), prevents catastrophic instability, but does not reduce the variance substantially. On the other hand, using a pathwise gradients is remarkably stable and exhibits small variance. However, it either requires access to the gradients of the objective function, or a strong surrogate model.

To use pathwise gradients in on-policy learning, our goal is thus to learn a suitable value function that allows us to estimate a low variance update direction without converging to a suboptimal solution.

## 3 Relative Entropy Pathwise Policy Optimization

Naively, we could take an off-policy algorithm like SAC and train it solely with data from the current policy. However, as Seo et al. (2025) recently showed, this can quickly lead to unstable learning.

To succeed in the on-policy regime, we need to be able to continually obtain new diverse data, and compute stable and reliable updates. Combining a set of recent advances in both reinforcement learning as well as neural network value function fitting, can satisfy these requirements. We first introduce the core RL algorithm, and then elaborate on the architectural design of the method.

At its core, REPPO proceeds similar to other on-policy actor-critic algorithms through three distinct phases: data gathering, value target estimation, and value and policy learning (see Algorithm 1). To obtain diverse data, REPPO uses a maximum-entropy formulation, adapted to multi-step TD-$\lambda$ (Subsection 3.1), to encourage exploration. Finally, to ensure that policies do not collapse and policy learning is stable, REPPO uses KL-constrained policy updates with a schedule that balances entropy-driven exploration and policy constraints (Subsection 3.2).

## 3.1 VALUE FUNCTION LEARNING

Off-policy PPG methods like TD3 (Fujimoto et al., 2018) and SAC (Haarnoja et al., 2018) mostly use single step Q learning, i.e. they use only immediate rewards for value function updates. This is paired with large replay buffers to stabilize learning. While on-policy algorithms cannot use past policy data, they can instead use low bias multi-step TD targets for stabilization (Fedus et al., 2020). Therefore, multi-step TD-$\lambda$ targets form the basis for our value learning objective. Note that REPPO is more closely related to SARSA than to Q-learning (Sutton & Barto, 2018), due to being on-policy.

In addition to multi-step returns, diverse data is crucial. To achieve a constant rate of exploration, and prevent the policy from prematurely collapsing to a deterministic function, we leverage the maximum entropy formulation for RL (Ziebart et al., 2008; Levine, 2018). The core aim of the maximum entropy framework is to keep the policy sufficiently stochastic by solving a modified policy objective which not only maximizes rewards but also penalizes the loss of entropy in the policy distribution. The maximum-entropy policy objective (Levine, 2018) can be defined as

$$J_{\text{ME}}(\pi_\theta) = \mathbb{E}_{\pi_\theta} \left[ \sum_{t=0}^{\infty} \gamma^t r(x_t, a_t) + \alpha \mathcal{H}[\pi_\theta(x_t)] \right], \tag{4}$$

where $\mathcal{H}[\pi_\theta(x)]$ is the entropy of the policy evaluated at $x$, and $\alpha$ is a hyperparameter which trades off reward maximization and entropy maximization. REPPO combines the maximum entropy objective with TD-$\lambda$ estimates, resulting in the following target estimate

$$G^{(n)}(x_t, a_t) = \sum_{k=t}^{n} \gamma^{k-t}(r(x_k, a_k) - \alpha \log \pi(a_k|x_k)) + \gamma^{n+1} Q(x_n, a_n) \tag{5}$$

$$G^\lambda(x, a) = \frac{1}{\sum_{n=0}^{N} \lambda^n} \sum_{n=0}^{N} \lambda^n G^{(n)}(x, a), \tag{6}$$

where $N$ is the maximum length of the future trajectory we obtain from the environment for the state-action pair $(x, a)$. Our implementation relies on the efficient backwards pass algorithm presented by Daley & Amato (2019). Crucially, the targets are computed on-policy after a new data batch is gathered, and the Q targets are not recomputed before gathering new data. Our Q learning loss is

$$\mathcal{L}_Q^{\text{REPPO}}\left(\phi|\{x_i, a_i\}_{i=1}^B\right) = \frac{1}{B} \sum_{i=1}^{B} \text{HL}\left[Q_\phi(x_i, a_i), G^\lambda(x_i, a_i)\right] + \mathcal{L}_{\text{aux}}(f_\phi(x_i, a_i), x_i'), \tag{7}$$

where $x_i'$ refers to the next state sample starting from $x_i$, and HL is the HL-Gauss loss (see Subsection 3.3 and Subsection D.2), and $\mathcal{L}_{\text{aux}}$ is presented in Subsection 3.3 and Subsection D.3.

Using purely on-policy targets allows us to remove several common off-policy stabilization components from the value learning setup. REPPO does not require a pessimism bias, so we can forgo the clipped double Q learning employed by many prior methods (Fujimoto et al., 2018). Tuning pessimistic updates carefully to allow for exploration is a difficult task (Moskovitz et al., 2021), so this simplification increases the robustness of our method. We also do not need a target value function copy, since we do not recompute the target at each step and it therefore remains on-policy.

## 3.2 POLICY LEARNING

A core problem with value-based on-policy optimization is controlling the size of the policy update, as the value estimate is only accurate on the data covered by the prior policy. A large policy update can therefore destabilize learning (Kakade & Langford, 2002). This problem has led to the development of constrained policy update schemes, where the updated policy is prevented from deviating too much from the behavioral (Peters et al., 2010; Schulman et al., 2015). To control the deviation, we use the Kullback-Leibler (KL) divergence, also called the relative entropy (Peters et al., 2010), as it can be justified theoretically through information geometry (Kakade, 2001; Peters & Schaal, 2008; Pajarinen et al., 2019), and is easy to approximate using samples.

Some works in the literature (Neumann, 2011; Sokota et al., 2022) claim that the reverse mode might be preferable for policy constraints, as it is mode-seeking, and the forward mode is mode-averaging. However, this intuition does not cleanly translate to our setting. As our policies are unimodal tanh-squashed Gaussian, the main impact of the KL direction is that the reverse-mode KL is entropy reducing. As we explicitly aim to increase the policy's entropy using the maximum entropy formulation, using forward-mode KL makes the optimization more stable.

**Policy Optimization Objective**   Our policy updates derive from a constrained optimization problem which includes both entropy and the KL constraint, and where $\theta'$ is the behavior policy, and $\varepsilon_{\mathrm{KL}}$ and $\varepsilon_{\mathcal{H}}$ are the respective KL and entropy constraints

$$\max_{\theta} \quad \mathbb{E}_{x \sim \rho_{\pi_{\theta'}}} \left[ \mathbb{E}_{a \sim \pi_{\theta}(\cdot|x)} \left[ Q(x, a) \right] \right] \tag{8}$$

$$\text{subject to} \quad \mathbb{E}_{x \sim \rho_{\pi_{\theta'}}} \left[ D_{\mathrm{KL}} \left( \pi_{\theta'}(\cdot|x) \,\|\, \pi_{\theta}(\cdot|x) \right) \right] \leq \varepsilon_{\mathrm{KL}} \tag{9}$$

$$\mathbb{E}_{x \sim \rho_{\pi_{\theta'}}} \left[ \mathcal{H}[\pi_{\theta}(\cdot|x)] \right] \geq \varepsilon_{\mathcal{H}}. \tag{10}$$

A similar combination of maximum entropy and KL divergence bound has been explored in various forms (Abdolmaleki et al., 2015; Pajarinen et al., 2019; Akrour et al., 2019). However, while previous approaches use complex solutions to this problem, such as approximate mirror descent, line search, or heuristic clipping, we take a simpler approach. We relax the problem, which introduces two hyperparameters, $\alpha$ for the entropy, and $\beta$ for the KL. Inspired by Haarnoja et al. (2019), REPPO automatically adapts these constraints when the policy violates them.

**Policy Updates and Multiplier Tuning**   In the constrained objective, we introduce two hyperparameters, $\varepsilon_{\mathcal{H}}$ and $\varepsilon_{\mathrm{KL}}$, which bound the entropy and KL divergence. The goal of the Lagrangian parameters is to ensure that the policy stays close to these constraints. As we need to ensure that they remain positive, we update them in log space with a gradient based root finding procedure

$$\alpha \leftarrow \alpha - \eta_{\alpha} e^{\alpha} \mathbb{E}_{x \sim \rho_{\pi_{\theta'}}} \left[ (\mathcal{H}[\pi_{\theta}(\cdot|x)] - \varepsilon_{\mathcal{H}}) \right] \tag{11}$$

$$\beta \leftarrow \beta - \eta_{\beta} e^{\beta} \mathbb{E}_{x \sim \rho_{\pi_{\theta'}}} \left[ (D_{\mathrm{KL}}(\pi_{\theta'}(\cdot|x) \| \pi_{\theta}(\cdot|x)) - \varepsilon_{\mathrm{KL}}) \right]. \tag{12}$$

Finally, to ensure our KL constraint is (approximately) maintained, we clip the actor loss based on whether the constrained is currently violated. The full policy objective for REPPO is now

$$\mathcal{L}_{\pi}^{\mathrm{REPPO}}(\theta|x_i) = \begin{cases} -Q(x_i, a) + e^{\alpha} \log \pi_{\theta}(a|x_i), & \text{if } \frac{1}{k} \sum_{j=1}^{k} \log \frac{\pi_{\theta'}(a_j|x_i)}{\pi_{\theta}(a_j|x_i)} < \varepsilon_{\mathrm{KL}} \\ e^{\beta} \frac{1}{k} \sum_{j=1}^{k} \log \frac{\pi_{\theta'}(a_j|x_i)}{\pi_{\theta}(a_j|x_i)}, & \text{otherwise} \end{cases} \tag{13}$$

where $a$ is sampled from $\pi_{\theta}(\cdot|x_i)$ and $a_j$ from the past behavior policy $\pi_{\theta'}(\cdot|x_i)$, and $k$ denotes how many samples are used to approximate the KL. As with the critic, the optimized loss is a mean over a minibatch from the rollout data. Note that contrary to other on-policy algorithms like PPO and TRPO, we are not forced to use actions sampled from the behavior policy in the policy gradient estimator, which removes the need for importance sampling correction. We will show that this greatly improves the performance of REPPO in Subsection 4.1.

Jointly tuning the entropy and KL multipliers is a crucial component of REPPO. As the policy entropy and KL are tied, letting the entropy of the behavior policy collapse means the KL constraint prevents any policy updates. Furthermore, the entropy and KL terms are balanced against the scale of the returns in the maximum entropy formulation. As the returns increase, keeping the multipliers fixed causes the model to ignore the constraints over time, accelerating collapse. However, as we tune both in tandem, our setup ensures a steady, measured update rate of the policy.

## 3.3 STABLE REPRESENTATION AND VALUE FUNCTION ARCHITECTURES

While the RL algorithm offers a strong foundation to obtain strong surrogate values, we also draw on recent off-policy advances in value function learning that improve training through architecture and loss design. We incorporate three major advancements into REPPO to further stabilize training.

**Cross-entropy loss for regression**   The first choice is to replace the mean squared error in the critic update with a more robust cross-entropy based loss function. For this, REPPO uses the HL-Gauss loss (Farebrother et al., 2024). This technique was adapted from the distributional C51 algorithm (Bellemare et al., 2017), which can lead to stable learning algorithms. Inspired by this insight and histogram losses for regression (Imani & White, 2018), Farebrother et al. (2024) hypothesize that observed benefits are due to the fact that many distributional algorithms use a cross-entropy loss, which is scale invariant. Palenicek et al. (2025) further investigate and reinforce this claim, showing that stable gradients arise from cross-entropy based losses. We present the mathematical form of the loss formulation in Subsection D.2. We find that a categorical loss is a crucial addition, as our ablation experiments show (Subsection E.1), but alternatives like C51 could easily work as well.

**Layer Normalization**   Several recent works (Ball et al., 2023; Yue et al., 2023; Lyle et al., 2024; Nauman et al., 2024a; Hussing et al., 2024; Gallici et al., 2024) have shown the importance of layer normalization (Ba et al., 2016) for stable critic learning. Gallici et al. (2024) provides a thorough theoretical analysis of the importance of normalization in on-policy learning, while Hussing et al. (2024) focuses on assessing the empirical behavior of networks in off-policy learning with and without normalization. As we operate in an on-policy regime where value function targets are more stable, we find that normalization is not as critical for REPPO as it is for off-policy bootstrapped methods; yet, we still see performance benefits in most environments from normalization.

**Auxiliary tasks**   Auxiliary tasks (Jaderberg et al., 2017) can stabilize features in environments with sparse rewards, where the lack of a reward signal can prevent learning meaningful representations via the Q learning objective (Voelcker et al., 2024a). For REPPO, auxiliary tasks are especially impactful when we decrease the number of samples used in each update batch (see Subsection E.1). We provide a discussion of this auxiliary task setup, including the loss function, in Subsection D.3.

## 4 EXPERIMENTAL EVALUATION

We begin by evaluating whether pathwise estimators improve upon score-based estimation in on-policy RL settings. We then compare our approach to baselines, evaluating final performance, sample and wall-clock efficiency, and stability of policy improvement. Our results demonstrate strong performance of REPPO on all axes. Additional details on architectures, hyperparameters, and ablations are provided in Subsection D.4 and Appendix E. A discrete variant of REPPO, along with its architectural changes and experimental results, is presented in Appendix C.

**Environments**   We evaluate REPPO on two major GPU-parallelized benchmark suites: 23 tasks from the mujoco_playground DMC suite (Zakka et al., 2025) and 8 ManiSkill environments (Tao et al., 2025), covering locomotion and manipulation, respectively. These tasks span high-dimensional control, sparse rewards, and chaotic dynamics.

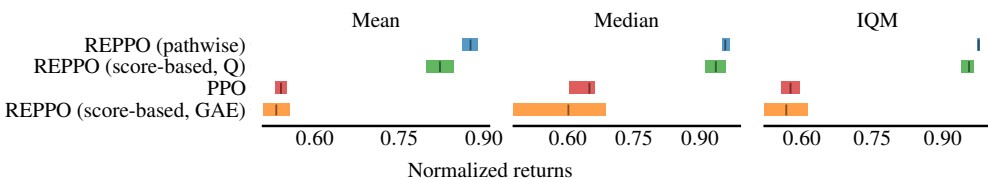

Figure 4: **Aggregate performance mujoco_playground** We compare REPPO with two ablations: using the score-based gradient estimator with the learned Q function, and using an on-policy GAE estimate with importance sampling and clipping. For additional context, we also report PPO results.

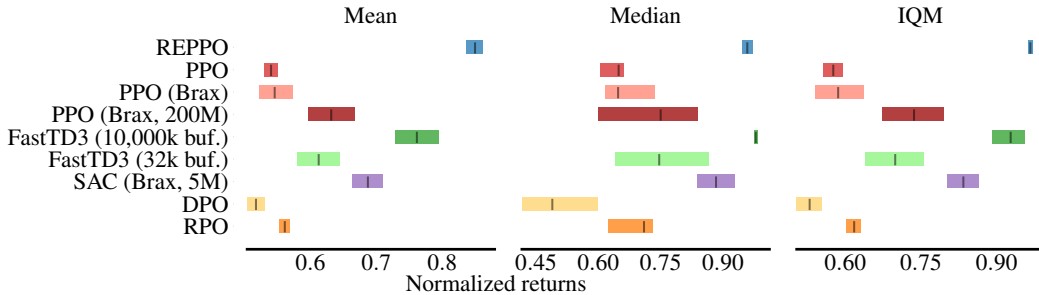

(a) **Aggregate performance mujoco_playground**. We compare REPPO to a re-tuned PPO baseline, the Brax PPO and SAC implementations provided by Zakka et al. (2025), as well as FastTD3 (Seo et al., 2025), RPO (Rahman & Xue, 2023), and DPO (Lu et al., 2022).

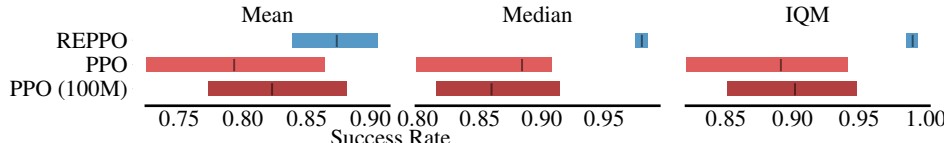

(b) **Aggregate success maniskill (Tao et al., 2025)**. We compare REPPO against a PPO baseline provided by Tao et al. (2025) at 50 million environment steps. As some environments take more than 50 million steps for PPO to achieve strong performance, we report the final performance at 100 million steps. While the mean confidence intervals are very broad, REPPO performs strongly on the IQM and median metrics.

Figure 5: Aggregate performance comparison on (a) mujoco_playground DMC and (b) ManiSkill3.

## 4.1 SCORE-BASED AND PATHWISE COMPARISON

REPPO offers an alternative to score-based policy gradient estimation in on-policy RL. However, we also introduce several enhancements, including automated tuning of entropy and KL coefficients, to improve value and policy learning. To assess the benefits of learned values and pathwise gradient estimation over score-based methods, we conduct two experiments. First, we replace the pathwise term $-Q(x, a)$ in Equation 13 with the score function $\log \pi(a|x)[Q(x, a)]_{\text{sg}}$, denoted as *REPPO (score-based, Q)*. Second, we replace the gradient estimator with the GAE-based clipped objective from PPO, denoted as *REPPO (score-based, GAE)*. Aggregate results are presented in Figure 4.

Using the approximate Q function in the policy gradient objective provides a strong improvement over a clipped objective. This showcases the benefits of value function learning and removing importance sampling. This also shows that the REPPO framework can be used with policy classes that are not amenable to reparameterization, such as diffusion policies (Chi et al., 2024; Celik et al., 2025; Ma et al., 2025), by using a score-based estimator together with the learned Q function. Interestingly, combining the PPO objective with REPPO leads to slightly worse results than vanilla PPO. We find that the high variance complicates the automatic parameter tuning scheme.

## 4.2 BENCHMARK COMPARISON

We compare REPPO against the PPO and SAC results reported by Zakka et al. (2025) and Tao et al. (2025). We report PPO baselines at 50M environment steps, and at the larger training horizon used in the original papers (Zakka et al., 2025). Results taken from Zakka et al. (2025) are denoted as "PPO/SAC (Brax)". To ensure that PPO is not undertuned for the 50m step regime we re-tuned the hyperparameters of the implementation provided by Lu et al. (2022). SAC results are reported at 5m steps as this amounts to similar total runtime as the 200m PPO results (compare results in Zakka et al. (2025). Naively running SAC at a larger sample budget and wall-clock efficiency can lead to instability, as Seo et al. (2025) demonstrates. Furthermore, we include FastTD3 (Seo et al., 2025) on DMC locomotion tasks, trained under two memory budgets: the default replay buffer (10,485,760 transitions) and a constrained buffer similar in size to on-policy methods (32,768 transitions) to control for the the memory and performance trade-off. Finally, we compare against

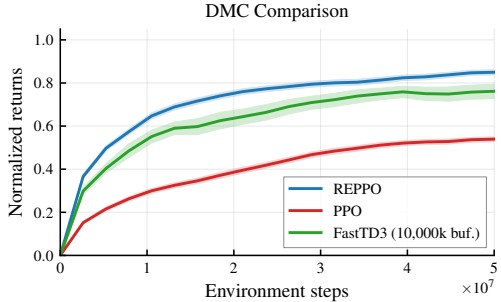
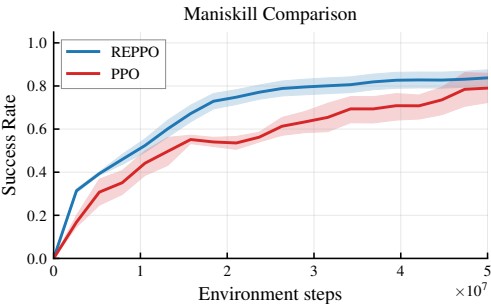

Figure 6: Aggregate sample efficiency curves for the benchmark environments. Settings are identical to those in Figure 5. REPPO achieves higher performance at a faster rate in both benchmarks.

Robust Policy Optimization (RPO) (Rahman & Xue, 2023) and Discovered Policy Optimization (DPO) (Lu et al., 2022). However, even with some hyperparameter tuning, we were unable to achieve a strong performance improvement beyond the PPO baseline with these approaches.

For REPPO, we report results aggregated over 20 seeds. We run 20 seeds for PPO and 5 for FastTD3[3], reporting aggregate scores with 95% bootstrapped confidence intervals (Agarwal et al., 2021). To aggregate scores, mujoco_playground returns are normalized by the maximum achieved by any algorithm. For ManiSkill we report raw success rates, which are comparable across tasks.

**Final Performance and Sample Efficiency** We first investigate the performance of policies trained using REPPO. We report aggregate performance at the end of training on both benchmarks in Figure 5. For both benchmarks, we also provide the corresponding training curves in Figure 6.

The aggregate results shown in Figure 5 and Figure 6 indicate that our proposed method achieves statistically significant performance improvements over PPO, as well as similar performance to FastTD3 despite REPPO being fully on-policy. Although these results are most pronounced in locomotion tasks, ManiSkill manipulation results show significant performance benefits over PPO in terms of outlier-robust metrics (Chan et al., 2020a; Agarwal et al., 2021).

We find that PPO struggles on high-dimensional tasks such as HumanoidRun. Moreover, despite its approximate trust-region updates, PPO suffers from performance drops and unstable training. This erratic behavior closely mirrors the score-based policy gradient instability shown in Figure 3a. In contrast, REPPO exhibits more stable improvements and lower variance across seeds.

**Wall-clock Time** Wall-clock time is an important metric, as it reflects the practical utility of an algorithm: faster training enables more efficient hyperparameter search and experimentation. However, measuring wall-clock time is nuanced, as results heavily depend on implementation details and are difficult to reproduce. We discuss these challenges across different frameworks in Appendix B. In Figure 7, we compare the wall-clock performance of our approach against PPO and SAC in JAX. Other baselines lack JIT-compilable implementations, making direct comparisons less meaningful.

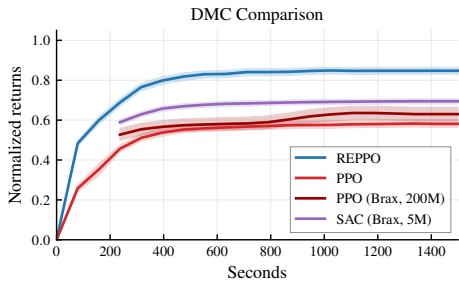

Figure 7: Wall-clock time comparison of REPPO against PPO and SAC implementations in JAX. REPPO matches other algorithms' speed but achieves higher return.

The computational cost per update is higher for REPPO than for PPO due to larger default networks and gradient propagation through the critic–actor chain. Nevertheless, both algorithms converge on most tasks in roughly 600–800 seconds, with REPPO achieving about 33% higher normalized returns. This shows that the sample efficiency

---

[3]We use fewer seeds for FastTD3 as we are unable to replicate the speed claimed in the paper. This is due pytorch specific issues discussed in Appendix B, and because we use smaller GPUs for our experiments.

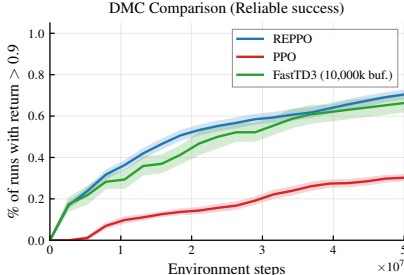 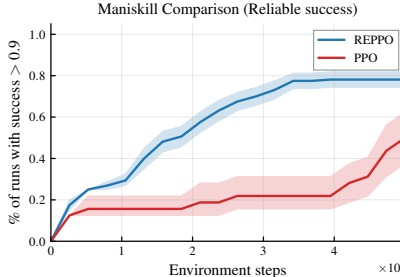

Figure 8: Fraction of runs that achieve reliable performance as measured by our metric for policy stability and reliability. REPPO's immediately starts achieving high performance in some runs and the number gradually increases indicating stable learning. PPO struggles to achieve high performance initially and to maintain high performance throughout training.

of pathwise gradients can offset their higher per-update cost, yielding improved wall-clock efficiency compared to score-based PPO. In addition, we find that jax-based SAC, which is tuned to trade sample for computational efficiency, slightly outperforms PPO, but does not match REPPO in performance. We note that other, modern SAC implementations (Nauman et al., 2024b; Lee et al., 2025a;b), are able to achieve better performance, but at the cost of computational efficiency.

**Reliable Policy Success**   We further investigate the stability of policy improvements using score-based and pathwise policy gradients. Our guiding principle is that such updates should not cause large drops in performance. To capture this, we adopt the "reliable success" metric, as proposed in Chan et al. (2020b). We define an algorithm as *reliably performant* if, once its performance exceeds a fixed threshold $\tau$, it never drops below this threshold thereafter. At each timestep, we track the number of runs that satisfy this criterion. This metric reflects the practical requirement that a deployed algorithm should not suddenly degrade simply due to continued training. We report the percentage of reliably successful runs for both REPPO and PPO in Figure 8.

On both DMC and ManiSkill benchmarks, REPPO achieves reliable performance improvements quickly. By the end of training, about four out of five runs have reached the threshold of $\tau = 0.9$, whereas PPO achieves roughly 40 percentage points fewer reliably performant runs. We also find notable differences in sample efficiency: PPO requires 5–10 million interactions before most envs become reliably performant. Overall, these results show that, despite relying on a biased value model, pathwise policy gradients enable stable long-term improvement.

## 5   CONCLUSION AND AVENUES FOR FUTURE WORK

In this paper we present REPPO, a highly performant yet efficient on-policy algorithm that leverages trained state-action value functions and pathwise policy gradients. By balancing entropic exploration and KL-constraints, and incorporating recent advances in neural network value function learning, REPPO is able to learn a high-quality approximation sufficient for reliable gradient estimation. As a result, the algorithm outperforms PPO on two GPU-parallelized benchmarks in terms of final return, sample efficiency and reliability while being on par in terms of wall-clock time. In addition, the algorithm does not require storing large amount of data making it competitive with recent advances in off-policy RL while requiring orders of magnitude lower amounts of memory.

As our method opens a new area for algorithmic development, it leaves open many exciting avenues for future work. As Seo et al. (2025) shows, using replay buffers can be beneficial to stabilize learning as well. This raises the question if our Q learning objective can be expanded to use both on- and off-policy data to maximize performance while minimizing memory requirements. Furthermore, the wide literature on improvements on PPO, such as learned objectives (Lu et al., 2022) can be incorporated into REPPO. We also observe that removing the importance sampling step in PPO has a crucial impact on performance, which suggests further research on the trade-off between efficiency and stability in on-policy gradient estimation is needed. Finally, better architectures such as Nauman et al. (2024b), Lee et al. (2025a), Otto et al. (2021) might be transferable to our algorithm and the rich literature on architectural improvements in off-policy RL can be tested in on-policy value learning.

## ACKNOWLEDGMENTS

CV acknowledges funding through the Ontario Graduate Scholarship. AMF acknowledges the support of NSERC through the Discovery Grant program [2021-03701], Polytechnique Montréal's PIED program, and IVADO IAR[3] grant. Resources used in preparing this research were provided, in part, by the Province of Ontario, the Government of Canada through CIFAR, and companies sponsoring the Vector Institute. EE and MH's research was partially supported by the DARPA Triage Challenge under award HR00112420305. Any opinions, findings, and conclusion or recommendations expressed in this material are those of the authors and do not necessarily reflect the view of DARPA or the US government.

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

## A    EXTENDED RELATED WORK

**Stabilizing On-Policy RL**    A fundamental issue with score-based approaches is their instability. Therefore, various improvements to decrease gradient variance have been considered. Some works have noted the difficulty of representation learning and have addressed this via decoupling the training of value and policy (Cobbe et al., 2021; Aitchison & Sweetser, 2022). Moalla et al. (2024) note that feature learning problems can result from representation collapse, which can be mitigated using auxiliary losses. There are also efforts to reduce the variance of gradients, e.g. by finding a policy that minimizes the variance of the importance sampling factor (Papini et al., 2024) or modifying the loss to ensure tighter total variational distance constraints (Xie et al., 2025).

Incorporating ground-truth gradient signal to stabilize training has also been studied, both for dynamical systems (Son et al., 2023) and differentiable robotics simulation (Mora et al., 2021; Xu et al., 2022; Georgiev et al., 2024). However, access to a ground-truth gradient requires custom simulators, and in contact-rich tasks, approximate models can provide smoother gradients (Suh et al., 2022).

**Trust regions and constrained policy optimization**    Other approaches have used similar KL and trust region constraint as REPPO. Schulman et al. (2015) and Peters et al. (2010) formulate the KL constrained policy update as a constrained optimization problem. Peters et al. (2010) shows a closed form solution to this problem, while Schulman et al. (2015) uses a conjugate gradient scheme to solve the relaxed optimization problem. Schulman et al. (2017) replaces the Lagrangian formulation with a clipping heuristic. However, clipping can lead to wrong gradient estimates (Ilyas et al., 2020) and in some scenarios the clipping objective fails to bound the policy deviation (Wang et al., 2020). Akrour et al. (2019) propose to project the policy onto the trust-region to sidestep the difficulty associated with clipping. We find that our approach is simpler to implement and more general, as we do not assume direct projection is possible.

Otto et al. (2021) propose to replace the various trust-region enforcement methods such as line-search or clipping with differentiable trust-region layers in the policy neural network architecture. While our method is slightly more general, as we make no assumption on the form of the policy (aside from assuming gradient propagation through the sampling process is possible), trust-region layers could easily be combined with REPPO for appropriate policy parameterizations.

**Work on GPU-parallelized On-policy RL**    With the parallelization of many benchmarks on GPUs (Makoviychuk et al., 2021; Zakka et al., 2025; Tao et al., 2025), massively-parallel on-policy RL has become quite popular. While these environments provide simulation testbeds, algorithms trained in such environments have shown to transfer to real-robots, allowing us to train them in minutes rather than days (Rudin et al., 2022).

**Hybridizing Off-policy and On-policy RL methods**    Most closely to our work, Parallel Q Networks (PQN) (Gallici et al., 2024) was established by using standard discrete action-space off-policy techniques in the MPS setting. While our work shares several important features with this method, we find that our additional insights on KL regularization and tuning is crucial for adapting the concept to continuous action spaces. We also evaluate our approach on discrete action spaces (see Appendix C). While PQN performs slightly better, likely owing to tuned exploration techniques, we show that our method works robustly across both discrete *and* continuous action spaces.

Other methods, such as Parallel Q-Learning (Li et al., 2023) and FastTD3 (Seo et al., 2025) also attempt to use deterministic policy gradient algorithms in the MPS setting, but still remain off-policy. This has two major drawbacks compared to our work. The methods require very large replay buffers, which can either limit the speed if data needs to be stored in regular CPU memory, or require very large and expensive GPUs. In addition, the off-policy nature of these methods requires stabilizing techniques such as clipped double Q learning, which has been shown to prevent exploration.

**KL-based RL**    Finally, other works also build on top of the relative entropy policy search (Peters et al., 2010). Maximum A Posteriori Policy Optimization (MPO) (Abdolmaleki et al., 2018) and Variational MPO (Song et al., 2019) both leverage SAC style maximum entropy objectives and use KL constraints to prevent policy divergence. However, both methods use off-policy data together with importance sampling, which we forgo, do not tune the KL and entropy parameters, and crucially do not make use of the deterministic policy gradient.

Going beyond relative entropy, the KL-based constraint formulation has been generalized to include the class of mirror descent algorithms (Grudzien et al., 2022; Tomar et al., 2022). In addition, Lu et al. (2022) meta-learns a constraint to automatically discover novel RL algorithms. These advancements are largely orthogonal to our work and can be incorporated into REPPO in the future.

**Instability in Off-policy RL**  Our method furthermore adapts many design decisions from recent off-policy literature. Among these are layer normalizations, which have been studied by Nauman et al. (2024a); Hussing et al. (2024); Nauman et al. (2024b); Gallici et al. (2024), auxiliary tasks (Jaderberg et al., 2017; Schwarzer et al., 2021; 2023; Tang et al., 2023; Voelcker et al., 2024b; Ni et al., 2024), and HL-Gauss (Farebrother et al., 2024), variants of which have been used by Hafner et al. (2021); Hansen et al. (2024); Voelcker et al. (2025). Beyond these, there are several other works which investigate architectures for stable off-policy value learning, such as Nauman et al. (2024b); Lee et al. (2025a;b). A similar method to our KL regularization tuning objective has been used by (Nauman & Cygan, 2025) to build an exploratory optimistic actor. While the technique is very similar, we employ it in the context of the trust-region update, and show the importance of jointly tuning the entropy and KL parameters. Finally, there are several papers which investigate the impact of continual learning in off-policy reinforcement learning, including issues such as out-of-distribution misgeneralization (Voelcker et al., 2025), plasticity loss (Nikishin et al., 2022; D'Oro et al., 2023; Lyle et al., 2023; Abbas et al., 2023). Since many of these works focus specifically on improving issues inherent in the off-policy setting, we did not evaluate all of these changes in REPPO. However, rigorously evaluating what network architectures and stabilization methods can help to further improve the online regime is an exciting avenue for future work.

## B  Wallclock Measurement Considerations

Measuring wall-clock time has become a popular way of highlighting the practical utility of an algorithm as it allows us to quickly deploy new models and iterate on ideas. Rigorous wall-clock time measurement is a difficult topic, as many factors impact the wall-clock time of an algorithm.

We chose to not compare the jax and torch versions head-to-head as we found significant runtime differences on different hardware, and the different compilation philosophies lead to different benefits and drawbacks. For example, jax' full jit-compilation trades a much larger initial overhead for significantly faster execution, which can amortize itself depending on the number of timesteps taken. This is the reason why we do not include FastTD3 in Figure 7, as only a PyTorch implementation of the algorithm exists. FastTD3 and REPPO use similar algorithms and hyperparameters, therefore, barring complexities like those discussed below, we expect them to perform at similar speeds.

More importantly, torch's compilation libraries are built to accelerate standard supervised and generative workflows, but do not support RL primitives equally well. As the CPU needs to load kernels during training which the GPU then executes, the CPU plays a much larger role in the speed measurements of the torch-based variant of REPPO. Especially the tanh-squashed log probability computation and the frequent resampling from the action space cannot be offloaded into an efficient kernel without providing one manually, which we have not done. This is likely due to the fact that torch keeps its random seed on the CPU. This is not a concern for jax, due to the fact that all kernels are statically compiled when the program is first executed, and random seeds are handled explicitly as part of the program state. Therefore, the CPU is under much lower load.

Instead of raw wall-clock time measurements, which can vary massively across framework and hardware, we recommend that the community treat the question of wall-clock time more carefully. While the actual time for an experiment can be of massive importance from a practical point of view, the advantages and limitations of current frameworks can obscure exciting directions for future work. For example REPPO is highly competitive with PPO when implemented in jax, but struggles somewhat in torch due to framework specific design choices.

## C  Discrete REPPO (D-REPPO)

One of the major advantages of PPO in the zoo of RL algorithms is the fact that it can be used in both continuous and discrete action settings. However, as we build on the DDPG/TD3/SAC line of work, the exposition of our algorithm has focused on the continuous setting alone.

Nonetheless, it is easy to adapt our approach to the discrete action setting as well. Following the proposal of Christodoulou (2019), we can circumvent the chained critic-actor gradient and compute the value of the current policy, the entropy, and the KL bound in closed form

$$\mathcal{L}_{\pi,\leq \mathrm{KL}}^{\mathrm{D-REPPO}}(\theta|B) = -\frac{1}{|B|} \sum_{i=1}^{|B|} \sum_{j=1}^{|A|} \pi_\theta(a_j|x_i) \left( Q(x_i, a_j) + e^\alpha \log \pi_\theta(a_j|x_i) \right) \tag{14}$$

$$\mathcal{L}_{\pi,> \mathrm{KL}}^{\mathrm{D-REPPO}}(\theta|B) = -\frac{1}{|B|} \sum_{i=1}^{|B|} e^\beta \sum_{j=1}^{|A|} \pi_{\theta'}(a_j|x_i) \log \frac{\pi_{\theta'}(a_j|x_i)}{\pi_\theta(a_j|x_i)} \tag{15}$$

$$\mathcal{L}_\pi^{\mathrm{D-REPPO}}(\theta|B) = \begin{cases} \mathcal{L}_{\pi,\leq \mathrm{KL}}^{\mathrm{D-REPPO}}(\theta|B), & \text{if } \sum_{j=1}^{k} \log \frac{\pi_{\theta'}(a_j|x_i)}{\pi_\theta(a_j|x_i)} < \varepsilon_{\mathrm{KL}} \\ \mathcal{L}_{\pi,> \mathrm{KL}}^{\mathrm{D-REPPO}}(\theta|B), & \text{otherwise.} \end{cases} \tag{16}$$

This variant of our algorithm still directly differentiates the full Q function objective, so can still be seen as a pathwise implementation. But computing the expectation in closed form circumvents the necessity to use a biased estimator for discrete sampling, such as the Gumbel-Softmax trick (Maddison et al., 2017; Jang et al., 2017; Fujimoto et al., 2024).

To investigate the benefits of our approach in the discrete action setting, we compare it against PQN (Gallici et al., 2024) and PPO. The main benefit of our approach over PQN is that it is a) a general algorithm that unifies both discrete and continuous action spaces, due to the underlying actor critic architecture, and b) that the principled entropy and KL objectives stabilize updates and encourages continuing exploration without an epsilon greedy exploration strategy.

We find that our algorithm is able to perform roughly on-par with PQN in the Atari-10 suite of games (cf. Table 1 and Figure 9) with only minor changes to the architecture to adapt to the Atari games benchmark. Notably, suitable settings for the KL and entropy target remain consistent even for the discrete action setting. We only find that the value of $\lambda = 0.65$ that is also recommended by Gallici et al. (2024) is superior to our default value of 0.95, likely due to the higher variance of the return in the atari games. While the high variance across Atari games makes drawing a clear conclusion difficult, we find that PQN seems to achieve slightly better performance. We find that this is most likely due to the fact that the algorithm adds explicit exploration noise, while we rely on the entropy and conservative KL terms to pace policy improvement.

Table 1: Aggregated Human-Normalized Atari-10 scores with 95% confidence intervals.

| Algorithm | Mean [CI] | Median [CI] | IQM [CI] |
|---|---|---|---|
| REPPO | 2.98 [2.64, 3.33] | 1.68 [1.48, 1.82] | 1.64 [1.54, 1.74] |
| PQN | 3.35 [3.00, 3.76] | 1.58 [1.48, 1.71] | 1.64 [1.58, 1.71] |

## D  IMPLEMENTATION DETAILS AND HYPERPARAMETERS

In the following, we present implementation details on experiments, as well as a hyperparameter overview.

### D.1  TOY EXAMPLE

To obtain the gradient descent comparison in Subsection 2.2 we used the 6-hump camel function, a standard benchmark in optimization. As our goal was not to show the difficulties of learning with multiple optima, which affect any gradient-based optimization procedure, but rather smoothness of convergence, we initialized all runs close to the global minimum. The surrogate functions were small three layer, 16 unit MLPs. To obtain a strong and a weak version, we used differing numbers of samples, visualized in Figure 10. Every algorithm was trained with five samples from the policy at every iteration. Finally, we tested several learning rates. We chose a learning rate which allows the ground-truth pathwise gradient to learn reliably. If a smaller gradient step size is chose, the Monte-Carlo estimator converges more reliably, at the cost of significant additional computation. We also

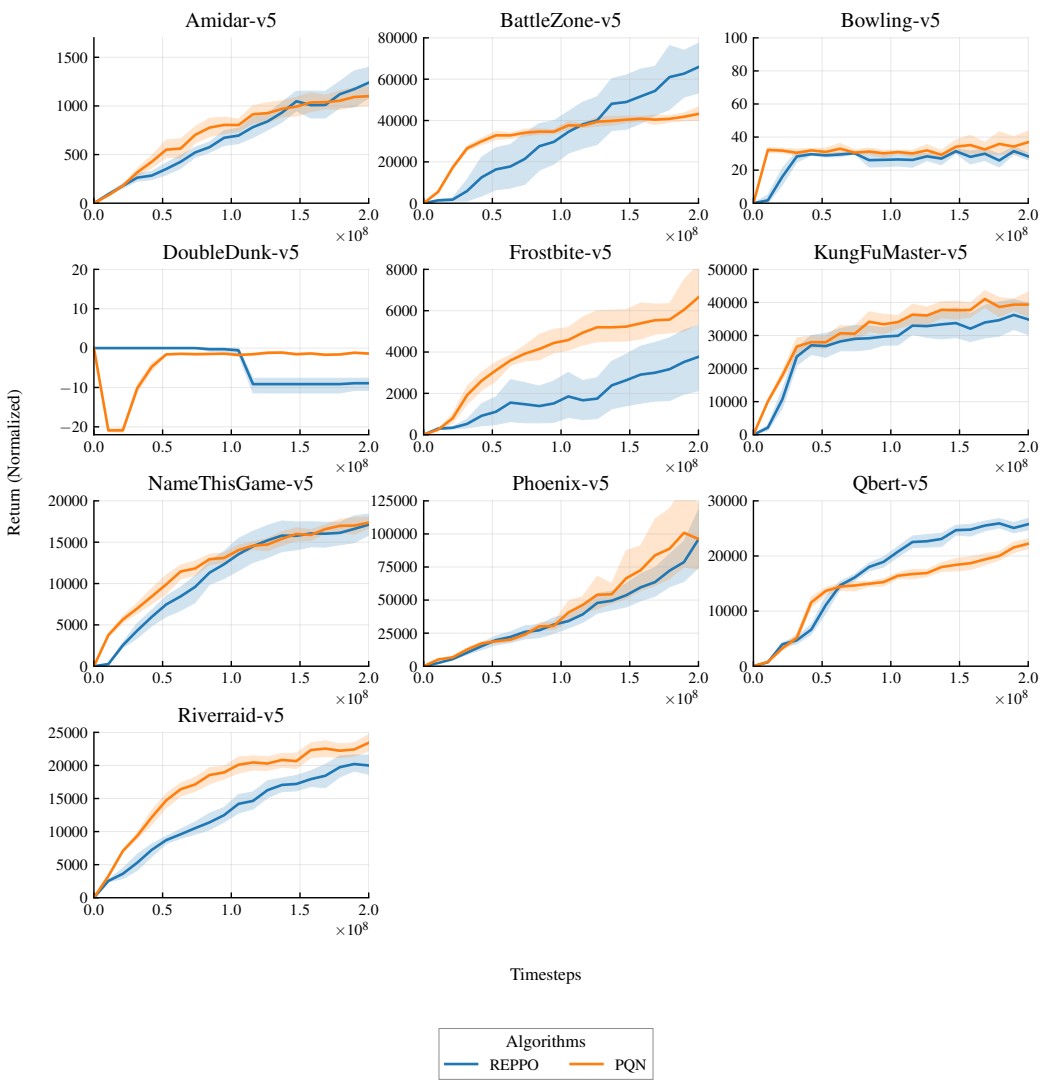

Figure 9: Per-environment results on the Atari-10 suite

tested subtracting a running average mean as a control variate from the Monte-Carlo estimate. While this reduced variance significantly, it was still very easy to destabilize the algorithm by choosing a larger step size or less data samples.

In total, our experiments further highlight a well known fact in gradient-based optimization: while a MC-based gradient algorithm can be tuned for strong performance, it is often extremely dependent on finding a very good set of hyperparameters. In contrast, pathwise estimators seem to work much more reliably across a wider range of hyperparameters, which corroborates our insights on REPPO hyperparameters robustly transfering across environemnts and benchmark suites.

## D.2 HL-GAUSS EQUATIONS

Given a regression target $y$ and a function approximation $f(x)$, HL-Gauss transforms the regression problem into a cross-entropy minimization. The regression target is reparameterized into a histogram approximation hist of $\mathcal{N}(y, \sigma)$, with a fixed $\sigma$ chosen heuristically. The number of histogram bins $h$ and minimum and maximum values are hyperparameters. Let $\text{hist}(y)_i$ be the probability value of the histogram at the $i$-th bucket. The function approximation has an $h$-dimensional output vector of

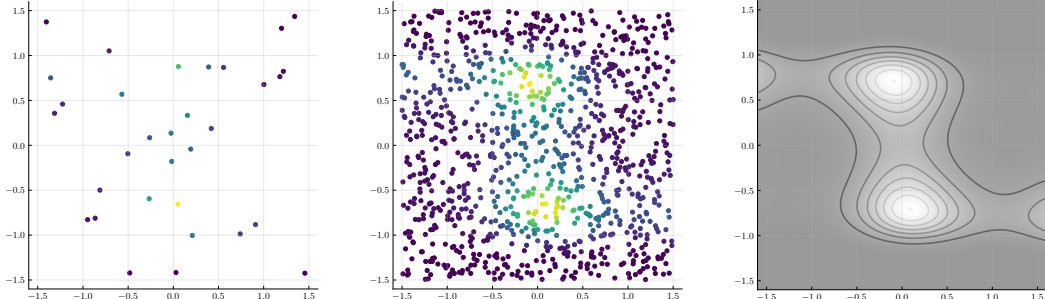

Figure 10: Samples used to train the surrogate function. On the left, we visualize the 32 sample dataset to train the weak surrogate function, in the middle the 1024 datapoints to train the strong, and on the right the full objective function.

logits. Then the loss function is

$$\mathrm{HL}(f(x), y) = \sum_{i=1}^{h} \mathrm{hist}(y)_i \cdot \log \frac{\exp f(x)_i}{\sum_{j=1}^{h} \exp f(x)_j} \ .$$

The continuous prediction can be recovered by evaluating

$$\hat{y} = \mathbb{E}[\mathrm{hist}(f(x))] = \langle \mathrm{hist}(f(x)), \mathrm{vec}(\min, \max, h) \rangle,$$

where $\mathrm{vec}(\min, \max, h)$ is a vector with the center values of each bin ranging from $\min$ to $\max$.

### D.3 AUXILIARY TASK SETUP

A simple yet impactful auxiliary task is latent self prediction (Schwarzer et al., 2021; Voelcker et al., 2024b; Fujimoto et al., 2024). In its simplest form, latent self-prediction is computed by separating the critic into an encoder $\phi : \mathcal{X} \times \mathcal{A} \to \mathcal{Z}$ and a prediction head $f_c : \mathcal{Z} \to \mathbb{R}$. The full critic can then be computed as $Q(x, a) = f_c(\phi(x, a))$. A self-predictive auxiliary loss adds a forward predictive model $f_p : \mathcal{Z} \to \mathcal{Z}$ and trains the encoder and forward model jointly to minimize

$$\mathcal{L}_{\mathrm{aux}}(x_t, a_t, x_{t+1}, a_{t+1}) = |f_p(\phi(x_t, a_t)) - \phi(x_{t+1}, a_{t+1})|^2 . \tag{17}$$

As our whole training is on-policy, we do not separate our encoder into a state-dependent and action dependent part as many prior off-policy works have done. Instead we compute the targets on-policy with the behavioral policy and minimize the auxiliary loss jointly with the critic loss.

Overall, the impact of the auxiliary task is the most varied across different environments. In some, it is crucial for learning, while having a detrimental effect in others. We conjecture that the additional learning objective helps retain information in the critic if the reward signal is not informative. In cases where the reward signal is sufficient and the policy gradient direction is easy to estimate, additional training objectives might hurt performance. We encourage practitioners to investigate whether their specific application domain and task benefits from the auxiliary loss.

### D.4 REPPO MAIN EXPERIMENTS

In addition to the details laid out in the main paper, we briefly introduce the architecture and additional design decisions, as well as default hyperparameter settings.

The architecture for both critic encoder and heads, as well as the actor, consists of several normalized linear layer blocks. As the activation function, we use silu/swift. As the optimizer, we use Adam. We experimented with weight decay and learning rate schedules, but found them to be harmful to performance. Hyperparameters are summarized in Table 2. We tune the discount factor $\gamma$ and the minimum and maximum values for the HL-Gauss representation automatically for each environment, similar to previous work (Hansen et al., 2024). This makes the hyperparameters, together with the algorithm description, and the source code, a *complete algorithm specification* in

| Environment | |
|---|---|
| total time steps | $50,000,000$ |
| n envs | $1024$ |
| n steps | $128$ |
| $\mathrm{KL_{tar}}$ | $0.1$ |
| Optimization | |
| n epochs | $8$ |
| n mini batches | $64$ |
| batch size | $\frac{\text{n envs} \times \text{n steps}}{\text{n mini batches}} = 2048$ |
| lr | $3e-4$ |
| maximum grad norm | $0.5$ |
| Problem Discount | |
| $\gamma$ | $1 - \frac{10}{\text{max env steps}}$ |
| $\lambda$ | $0.95$ |

| Critic Architecture | |
|---|---|
| critic hidden dim | $512$ |
| vmin | $\frac{1}{1-\gamma} \min r$ |
| vmax | $\frac{1}{1-\gamma} \max r$ |
| num HL-Gauss bins | $151$ |
| num critic encoder layers | $2$ |
| num critic head layers | $2$ |
| num critic pred layers | $2$ |
| Actor Architecture | |
| actor hidden dim | $512$ |
| num actor layers | $3$ |
| RL Loss | |
| $\beta$ start | $0.01$ |
| $\varepsilon_{\mathrm{KL}}$ | $0.1$ |
| $\alpha$ start | $0.01$ |
| $\varepsilon_{\mathcal{H}}$ | $0.5 \times \dim \mathcal{A}$ |
| aux loss mult | $1.0$ |

Table 2: Default REPPO hyperparameters

the sense of Jordan et al. (2020), as we only vary hyperparameters across environments following simple equations on clear, domain sepcific hyperparameters such as the size of the action space and the length of the experiment.

For all environments, we use observation normalization statistics computed as a simple running average of mean and standard deviation. We found this to be important for performance, similar as in other on policy algorithms. Since we do not hold data in a replay buffer, we do not need to account for environment normalization in a specialized manner, and can simply use an environment wrapper.

For more exact details on the architecture we refer to interested readers to the codebase.

# E ADDITIONAL RESULTS

In the following, we provide additional results and further clarification on existing experiments in Section 4.

## E.1 DESIGN ABLATIONS

We run ablation experiments investigating the impact of the design components used in REPPO. In these experiments, we remove the cross-entropy loss via HL-Gauss, layer normalization, the auxiliary self-predictive loss, or the KL regularization of the policy updates. To understand the importance of each component for on-policy learning we conduct these ablations for two scales of batch sizes - the default $131,072$ on-policy transitions, as well as the smaller batch size of $32,768$.

As shown in Figure 11, our results indicate that both the KL regularization of the policy updates and the categorical Q-learning via HL-Gauss are necessary to achieve strong performance independent of the size of the on-policy data used to update our model. We find that the KL divergence is the only component that, when removed, leads to a decrease in performance below the levels of PPO, which clarifies the central importance of relative entropy regularization for REPPO. Removing normalization has minor negative effects on performance which become worse at smaller buffer sizes. This is consistent with the literature on layer normalization in RL. Similarly, the auxiliary self-predictive loss has a more clearly negative impact on performance when the batch size becomes smaller. We note that auxiliary loss has an inconsistent impact on the training generally, where it is strongly beneficial in some environments, but harmful in others.

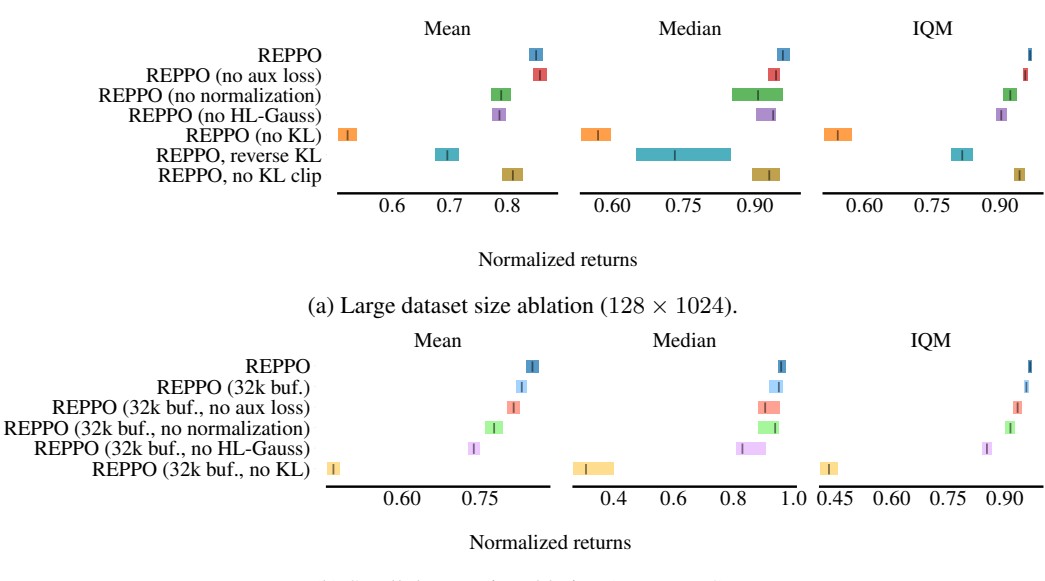

(a) Large dataset size ablation ($128 \times 1024$).

(b) Small dataset size ablation ($32 \times 1024$).

Figure 11: Ablation on components and data size on the DMC benchmark. Both values are significantly smaller than the replay buffer sizes used in standard off-policy RL algorithms like SAC and FastTD3. The HL-Gauss loss and KL regularization provide a clear benefit at both data scales. The normalization and auxiliary loss become more important when less data is available, highlighting that some stability problems can also be overcome with scaling data.

### E.2 MEMORY DEMANDS AND DATA SCALING

Recent advances in off-policy algorithms have shown great performance when large buffer sizes are available (Seo et al., 2025). When dealing with complex observations such as images, on-policy algorithms which do not require storing past data have a large advantage. In terms of data storage requirements, our algorithm is comparable with PPO, yet it remains to answer how well REPPO compares to algorithms that are allowed to store a large amount of data. For this, we compare against the recent FastTD3 (Seo et al., 2025) which also uses GPU-parallelized environments but operates off-policy. We compare REPPO against the original FastTD3 and we also re-run FastTD3 with access to a significantly smaller buffer equivalent to the REPPO buffer. We report the results in Figure 12b.

The results demonstrate that REPPO is on par or better in terms of performance on mean and IQM with the FastTD3 approach. This is despite the fact that REPPO uses a buffer that is two to three orders of magnitude smaller. When decreasing the buffer size of FastTD3, the algorithm's performance drops by a large margin while REPPO is barely affected by a smaller buffer. We find that FastTD3 with a smaller buffer can retain performance on lower dimensional, easier tasks but suffers on harder tasks that may be of greater interest in practice. In summary, REPPO is competetive with recent advances in off-policy learning with significantly lower memory and storage requirements.

These results raise the question whether positive scaling with replay buffer size is a general feature of on-policy algorithms. In our default configuration, REPPO uses long rollouts and a high number of parallel environments, as well as a large number of policy and value function update steps. PPO on the other hand is often implemented with smaller dataset sizes. We therefore set up REPPO and PPO training runs across 4 datasets, varying the rollout length. To keep the total number of gradient steps and the minibatch size the same, we reduced the number of minibatches proportionally to the batch size. The settings are summarized in Figure 12a. Note that in the large settings, the data becomes more off-policy. Both PPO and REPPO have explicit ways to deal with this, clipping and the KL minimization term respectively, but the clipping term in PPO is only a heuristic to prevent large importance sampling ratios.

| | Num envs | Num steps | Num minibatches | Epochs | Updates per batch |
|---|---|---|---|---|---|
| 130k buffer | 1024 | 128 | 64 | 8 | 512 |
| 65k buffer | 1024 | 32 | 16 | 8 | 128 |
| 32k buffer | 1024 | 8 | 4 | 8 | 32 |
| 16k buffer | 256 | 8 | 1 | 8 | 8 |

(a) Dataset configurations for the data scaling experiment.

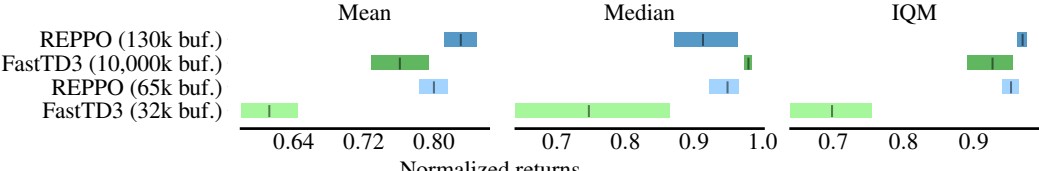

(b) Comparison of aggregate performance between REPPO and FastTD3. REPPO is competitive with the large buffer FastTD3 version and outperforms FastTD3 when memory is limited.

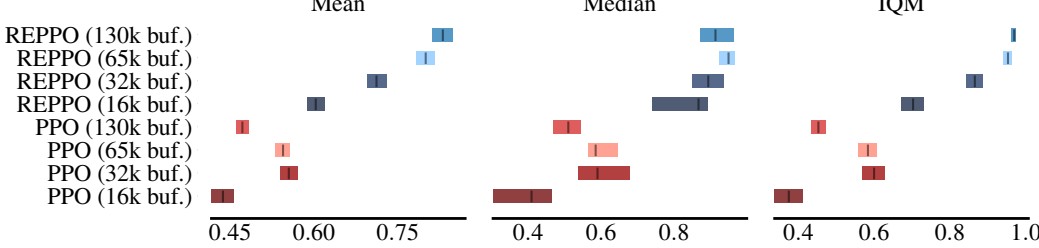

(c) Aggregated performance of REPPO and PPO under different batch dataset sizes. The mean performance of REPPO drops monotonically with decreasing batch size, while PPO shows its highest performance with a medium and small dataset size.

Figure 12: Experiment to compare the impact of batch datset size on different on-policy algorithms.

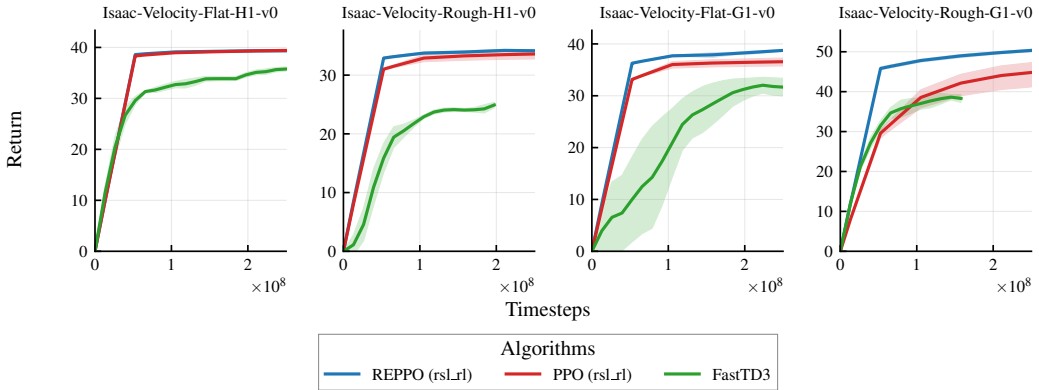

Figure 13: Result overview of REPPO on the IsaacLab locomotion tasks

Comparing the performance of both approaches (see Figure 12c), we observe a clear pattern. The mean performance of REPPO drops steeply with decreasing dataset size. PPO on the other hand does best in the medium and small dataset regimes. This highlights the different mechanisms on which both algorithms operate. Larger datasets allow the trained Q function to generalize better, similar to the insight presented in Figure 3a. On the other hand, for PPO the dataset size needs to be large enough to allow for stable gradient estimation, but not so large that too many gradient update steps are necessary. This is because clipping can prevent further learning, and many update steps can exacerbate varaince issues with importance sampling.

Note that at some point, REPPO will likely also stop improving with larger datasets and more gradient update steps. We see that the performance differences between the medium and the large dataset at not as strong as with smaller datasets. REPPO cannot continue to learn on fixed data forever, by design, as the KL divergence between two consecutive policies is constrained. However, we can hypothesize based on the empirical evidence that REPPO is able to scale more gracefully with large amounts of data.

Overall, we can conclude that REPPO scales favorably with larger data buffers, similar to SAC or FastTD3, but is not strongly reliant on it.

### E.3   ISAACLAB RESULTS

To provide additional value to the robotics community, who often favor specific implementations and APIs for reinforcement learning algorithms, we provide a reimplementation of REPPO in RSL-RL Schwarke et al. (2025). We ran this implementation of REPPO and compare the results on the humanoid locomotion environments in the Isaaclab suite Mittal et al. (2025). Results are presented in Figure 13. The FastTD3 results provided by Seo et al. (2025) are not all ran for the full 250 million timesteps.

For IsaacLab, we found that the KL constraint needs to be set more tightly to 0.01, which is the same value used for PPO.

### E.4   PER ENVIRONMENT SAMPLE EFFICIENCY CURVES

Finally, we provide sample efficiency curves per environment in Figure 14, Figure 15, and Figure 16.

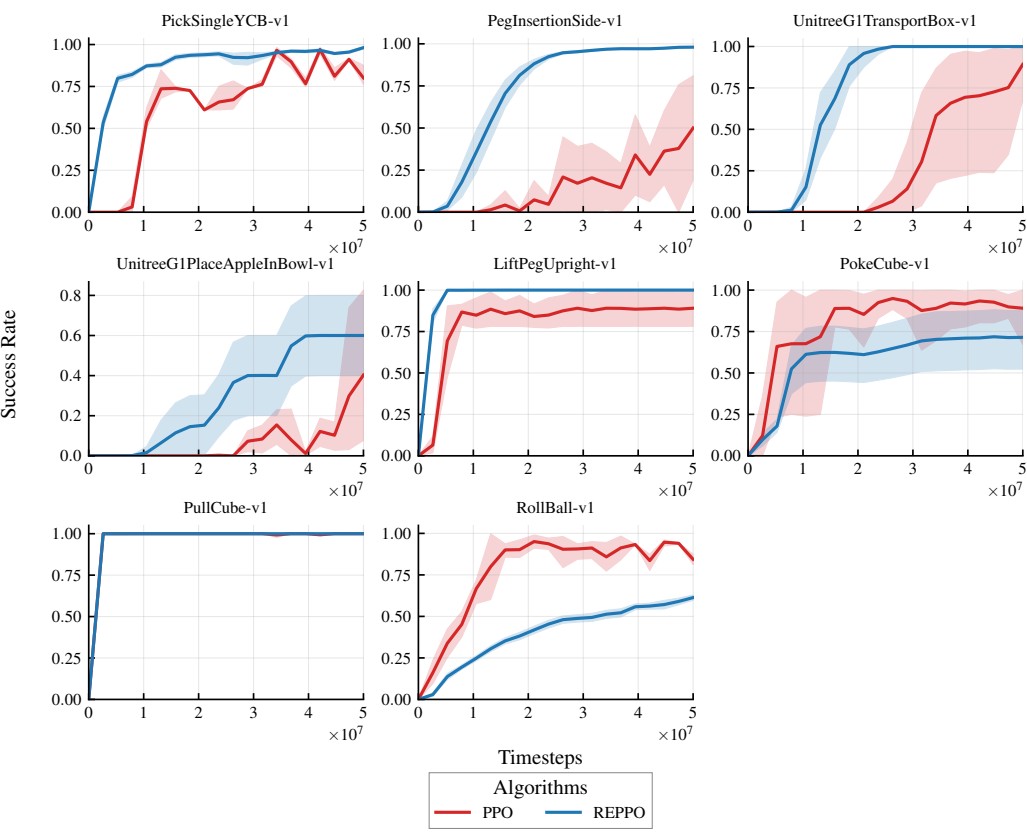

Figure 14: Per-environment results on the ManiSkill suite

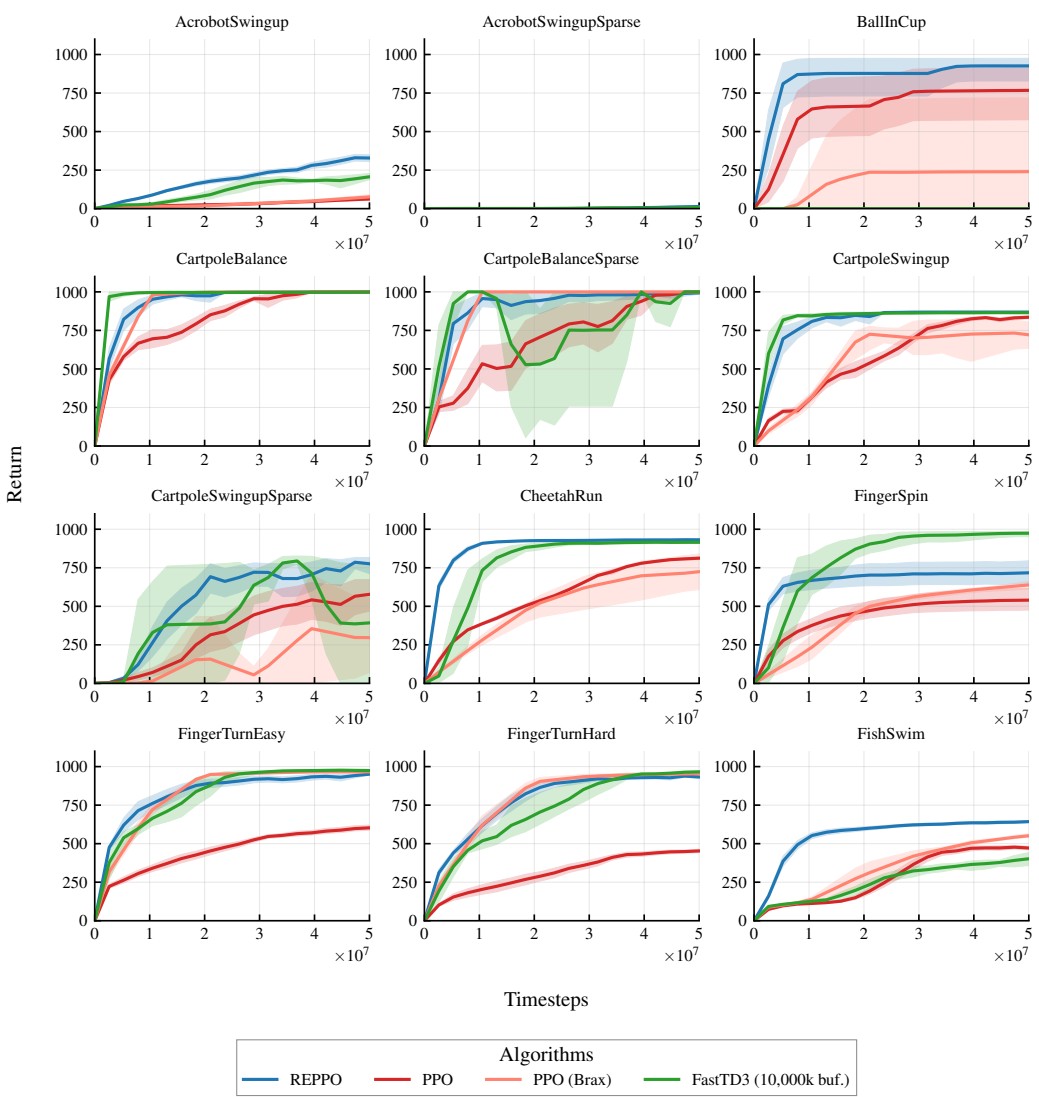

Figure 15: Per-environment results on the mujoco_playground DMC suite

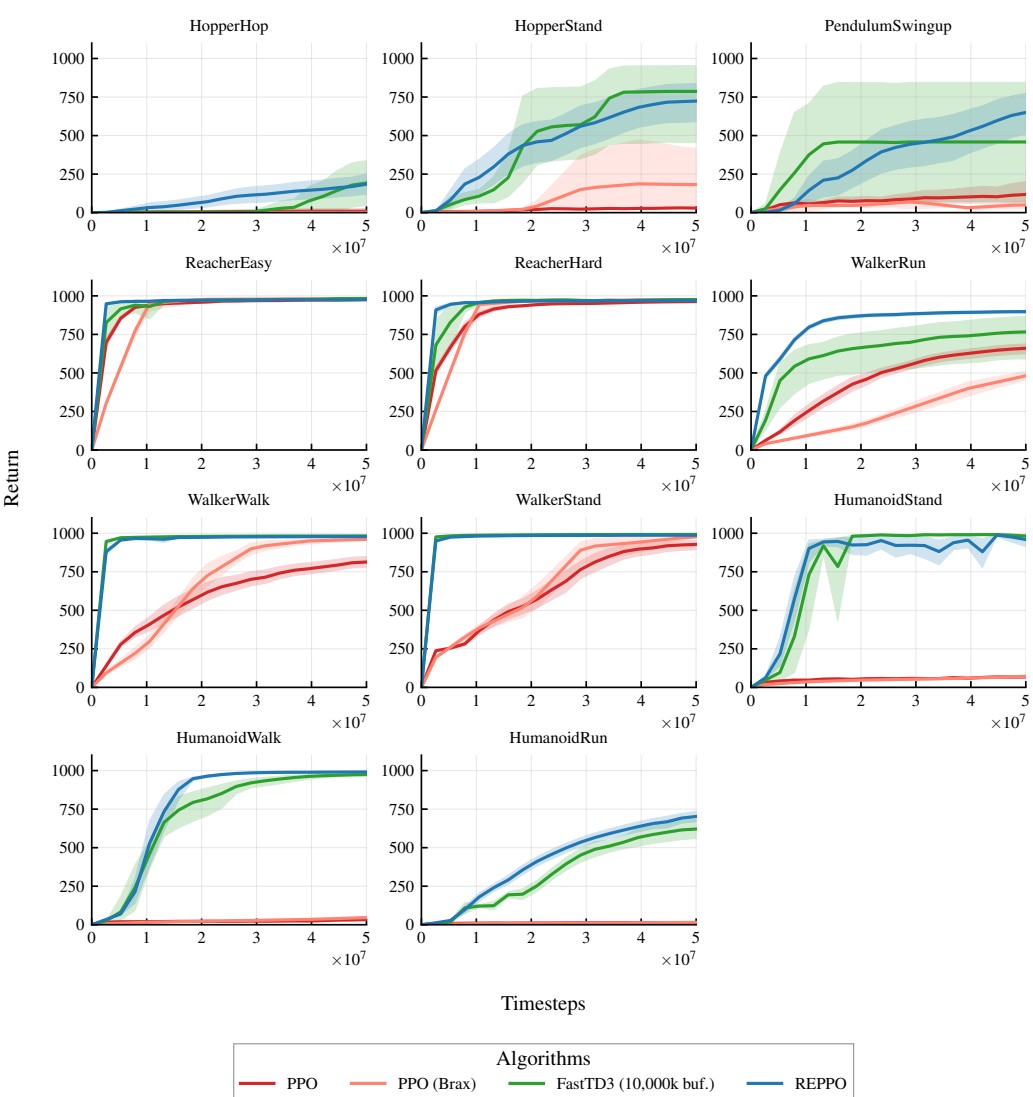

Figure 16: Per-environment results on the mujoco_playground DMC suite

# F  PSEUDOCODE

**Algorithm 1:** Pseudocode for Relative Entropy Pathwise Policy Optimization

**Input:** Environment $\mathcal{E}$, actor network $\pi_\theta$, critic network $Q_\phi$, hyperparameters
**Output:** Trained policy $\pi_\theta$
```
// Initialize networks
```
Actor $\pi_\theta$, behavior policy $\pi_{\theta'}$ with $\theta' = \theta$, critic $Q_\phi$ with encoder $f_\phi$, entropy and KL temperature $\alpha$ and $\beta$
**for** *iteration = 1 to $N_{iterations}$* **do**
    ```// Step 1:  Collect rollout with behavior policy```
    **for** *step = 1 to $N_{steps}$* **do**
        ```// Apply exploration noise scaling```
        Sample action $a_t \sim \pi_{\theta'}(\cdot|x_t)$
        Execute $a_t$ in environment, observe $(x_{t+1}, r_t, d_t)$
        Compute approximate $V_{t+1} \leftarrow Q_\phi(x_{t+1}, a_{t+1})$ with $a_{t+1} \sim \pi_{\theta'}(\cdot|x_{t+1})$
        Compute $\psi_t \leftarrow f_\phi(x_{t+1}, a_{t+1})$
        ```// Maximum entropy augmented reward, see``` Subsection 3.1
        $\tilde{r}_t \leftarrow r_t - \alpha \log \pi_\theta(a_{t+1}|x_{t+1})$
        Store transition $(x_t, a_t, \tilde{r}_t, x_{t+1}, d_t, V_{t+1}, \psi_t)$
    **end**
    ```// Step 2:  Compute TD-λ targets, see``` Subsection 3.1
    **for** *t = T − 1 down to 0* **do**
        $G_t^\lambda \leftarrow \tilde{r}_t + \gamma[(1 - d_t)(\lambda G_{t+1}^\lambda + (1 - \lambda)V_{t+1})]$
    **end**
    ```// Step 3:  Update networks for multiple epochs```
    **for** *epoch = 1 to $N_{epochs}$* **do**
        Shuffle data and create mini-batches
        **for** *each mini-batch $b = \{(x, a, G^\lambda, \psi)_i\}_{i=1}^B$* **do**
            ```// Categorical critic update, see``` Subsection 3.3
            $L_Q \leftarrow \frac{1}{B} \sum \text{CrossEntropy}(Q_\phi(x_i, a_i), \text{Cat}(G_i^\lambda))$
            ```// Auxiliary task, see``` Subsection 3.3
            $L_{aux} \leftarrow \frac{1}{B} \sum ||f_\phi(x_i, a_i) - \psi_i||^2]$
            Update critic: $\phi \leftarrow \phi - \alpha_Q \nabla_\phi(L_Q + \beta L_{aux})$
            ```// Actor update with entropy and KL regularization, see```
                    Subsection 3.1 ```and``` Subsection 3.2
            Sample action $a_i' \sim \pi_\theta(\cdot|x_i)$
            Sample k actions $\bar{a}_i \sim \pi_{\theta'}(\cdot|x_i)$
            Compute KL divergence: $D_{\text{KL}}(x_i) \leftarrow \sum_{j=1}^k \log \frac{\pi_{\theta'}(\bar{a}_j|x_i)}{\pi_\theta(\bar{a}_j|x_i)}$
            Policy loss: $L_\pi \leftarrow \frac{1}{B} \sum Q_\phi(x_i, a_i') - e^\alpha \log \pi_\theta(a_i'|x_i) - e^\beta D_{\text{KL}}(x_i)$
            (Alternatively, compute clipped objective)

            Update actor: $\theta \leftarrow \theta + \eta_\pi \nabla_\theta L_\pi$
            Entropy $\alpha$ update: $\alpha \leftarrow \alpha - \eta_\alpha \nabla_\alpha e^\alpha (\frac{1}{B} \sum \mathcal{H}[\pi_\theta(x_i)] - \varepsilon_\mathcal{H})$
            KL $\beta$ update: $\beta \leftarrow \beta - \eta_\beta \nabla_\beta e^\beta (\frac{1}{B} \sum D_{KL}(x_i)] - \varepsilon_{\text{KL}})$
        **end**
    **end**
    ```// Behavior Policy Update```
    $\theta' \leftarrow \theta$
**end**
**return** *Trained policy $\pi_\theta$*

