# OpenReview forum: "Relative Entropy Pathwise Policy Optimization"
_ICLR.cc/2026/Conference — ICLR 2026 Poster_

### Official Review · Reviewer_XoJ7 · 2025-10-21

**Soundness:** 3
**Presentation:** 4
**Contribution:** 4
**Rating:** 6
**Confidence:** 4

**Summary:**

The authors propose an on-policy algorithm that utilizes a pathwise gradient estimator with a learned Q-function, along with specific design choices to stabilize the proposed algorithm. The proposed algorithm only involves computing policy updates on on-policy actions, which avoids importance sampling and instability in the traditional PPO method.

**Strengths:**

- The design principle of the algorithm is, to my knowledge, novel, sound, and interesting
- The algorithm is more on-policy than traditional PPO, and therefore, more stable. The authors showcase the performance benefits in extensive benchmark tests
- Compared with off-policy methods, the proposed method does not require large memory consumption for a replay buffer
- The presentation is clear and flows smoothly

**Weaknesses:**

- I think a more thorough justification (theoretical and/or empirical) about what the main factors are that bring the claimed performance benefit can significantly strengthen the authors' claims; cf. my question about Figure 3 below. Also, if learning a Q function instead of V function alleviates the high-variance problem in PPO, what are the potential costs? I think a more careful justification will have longer-term benefits for the community and further development of this type of method
- Algorithm 1 (line 1364 of the document): V_{t+1} is not equal to Q_{\phi}(x_{t+1}, a_{t+1}) (missing an expectation over action)
- Policy optimization objective: the actual objective function used is not given in the text. Also, the notation used KL_{tar} is a bit arbitrary
- If some of the improtant baselines are indeed using different number of total samples (see my question below), then, I will suggest a proper comparison to those baselines with same access to the environment samples as REPPO
- Minor:
    - Line 246 typo “constraint it” -> “is”
    - Equation 12 is the gating reverse KL, whereas other places are using the forward KL divergence. Which one is true?
    - Current way of writing equation 12 is a bit off; consider pulling out the curly brace definition separately

**Questions:**

- In Figure 2: why does PPG with learned surrogate optimize better than the ground truth, even if the contour lines of the objective look quite simple and one would expect following the gradient should be the best direction? Also, what do you use as Q(s, a) for the score-based estimator in this case (should the score-based estimator also have different variants using different surrogates for Q(s, a))?

- Figure 3: The fact that the performance difference between REPPO (pathwise) and REPPO (score-based, Q) is tiny, and they both outperform PPO and REPPO (score-based, GAE) by a large margin, makes me wonder where the main performance gain of REPPO comes from. I am thinking that the performance benefit might mainly come from the fact that REPPO (pathwise) and REPPO (score-based, Q) query on-policy actions computed from the current policy (as opposed to the other two that use fixed, potentially off-policy actions sampled from data batches), which avoids the need for importance sampling scores. If that’s the case, then using a score-based vs. a pathwise estimator only makes a rather small difference. Please correct me if I am missing anything here.

- Figure 4: Why are many of the baseline results using a different number of environment steps? For example, is SAC using 5M steps, i.e., 1/10 of those used by REPPO? If that’s the case, I think it is not a fair comparison, and the result for SAC is not a meaningful reference. Therefore, the claim about statistically significant improvement over SAC is overclaiming as well.

---

> ### Author Response · Authors · 2025-11-17
> **Review reply**
>
> Dear reviewer,
>
> we are happy to answer the questions you raise.
> If there are any further improvements we can make, we are also happy to revisit and update the draft.
>
> > I think a more thorough justification (theoretical and/or empirical) about what the main factors are that bring the claimed performance benefit can significantly strengthen the authors' claims; cf. my question about Figure 3 below.
>
> > Figure 3: The fact that the performance difference between REPPO (pathwise) and REPPO (score-based, Q) is tiny, and they both outperform PPO and REPPO (score-based, GAE) by a large margin, makes me wonder where the main performance gain of REPPO comes from.
>
> We will answer both of these questions together, because we agree with the reviewer that this is an important point.
> Figure 3 shows a major benefit of the pathwise over reparameterization-based approach is that we can forgo the importance sampling corrections.
> However, this was not stressed strongly in our exposition.
> We have adapted the wording in the introduction to frame the importance of on-policy resampling more strongly.
> We also added an additional experiment to Section 2, using the same setup as before, to highlight the importance of resampling on-policy actions.
>
> > Also, if learning a Q function instead of V function alleviates the high-variance problem in PPO, what are the potential costs? I think a more careful justification will have longer-term benefits for the community and further development of this type of method
>
> The cost is that we are using a biased approximation of the policy gradient due to the surrogate Q function, while a vanilla REINFORCE style policy gradient would be unbiased (but suffer from high variance).
> This is also why we phrase our research question as "can we learn an accurate surrogate value function".
> We added additional language in the introduction to stress that our claim rests on this property.
> This means that practitioners will have to test whether the data and architectures they have available is indeed sufficient to use REPPO.
> This is a standard form of bias/variance trade-off, which is common to a large variety of approaches.
>
> Another, separate question, is whether there is a difference in sample complexity between learning a state and a state-action value function.
> We are unaware of any results of this kind, but it would be an interesting question to address in future work.
>
> > Algorithm 1 (line 1364 of the document): $V_{t+1}$ is not equal to $Q_{\phi}(x_{t+1}, a_{t+1})$ (missing an expectation over action)
>
> We have changed the notation to clarify that this is a sample based approximation.
>
> > Policy optimization objective: the actual objective function used is not given in the text. Also, the notation used $KL_{tar}$ is a bit arbitrary
>
> We have written out the policy objective in equations 7,8,9 more clearly. Please let us know if this adequately addresses your concern. We also replace $KL_{tar}$ with the less confusing $\epsilon_{tar}$.
>
> > If some of the important baselines are indeed using different number of total samples (see my question below), then, I will suggest a proper comparison to those baselines with same access to the environment samples as REPPO
>
> We evaluated two algorithms at different sample budgets: PPO at 200M samples (in addition to 50M) and SAC at 5M samples. For PPO, we adopted the 200M setting because the mujoco playground authors reported results at this scale. The 5M-sample SAC results, also taken from the mujoco playground paper, correspond to a similar wall-clock runtime as the 200M PPO runs. Consequently, both actually benefit from a larger compute budget than REPPO. To make this explicit, we added the SAC results to the wall-clock comparison plot.
>
> Results from FastTD3 give a good estimate for the performance of an off-policy algorithm that is tuned for a large sample regime, and as our results show, we perform roughly on par while requiring a significantly smaller memory footprint. The FastTD3 paper also shows that SAC specifically is unstable when naively adapted to the FastTD3 training regime, which is why we didn't attempt to run it at that scale.
>
> > Equation 12 is the gating reverse KL, whereas other places are using the forward KL divergence. Which one is true?
>
> You are correct that in some equations we have erroneously written out the reverse KL, thank you for catching this. We have corrected this inconsistency in the revised version.
> We also added a short paragraph that outlines our intuition behind using the forward KL.
> If the reviewer would like to see empirical evidence as well, we are happy to provide a comparison of forward and reverse KL.
>
> > Current way of writing equation 12 is a bit off; consider pulling out the curly brace definition separately
>
> We have simplified the notation by writing the loss per sample and noting that we simply average over a minibatch for the full loss.

---

> > ### Author Response · Authors · 2025-11-17
> > **Reply Part 2**
> >
> > ## Questions
> >
> > > In Figure 2: why does PPG with learned surrogate optimize better than the ground truth, even if the contour lines of the objective look quite simple and one would expect following the gradient should be the best direction? Also, what do you use as Q(s, a) for the score-based estimator in this case (should the score-based estimator also have different variants using different surrogates for Q(s, a))?
> >
> > The improved performance of the surrogate gradient estimator is indeed a curiosity. We did not expect this phenomenon to occur and debated changing the figure, as it is likely function and starting point dependent. However this was the first setup we tried and so we did not feel it would be a good idea to keep searching configurations until the figure looked the way we expected it to. Similar phenomena can be found in the literature on comparing surrogate approximations to differentiable simulators [1]. Surrogate functions can be smoother than the original, which can lead to slightly accelerated convergence of gradient based approaches. However, we do not believe that surrogate functions will lead to faster convergence in general.
> >
> > If the reviewers believe that this figure is too confusing or potentially misleading we are happy to update it.
> >
> > For the score-based estimator we used the ground truth objective function to ensure that it is unbiased. We did not use additional surrogates as we are only dealing with a simple 2d objective function and not a full RL problem.
> >
> > > Figure 4: Why are many of the baseline results using a different number of environment steps? For example, is SAC using 5M steps, i.e., 1/10 of those used by REPPO? If that’s the case, I think it is not a fair comparison, and the result for SAC is not a meaningful reference. Therefore, the claim about statistically significant improvement over SAC is overclaiming as well.
> >
> > We clarified the basis of comparison with SAC above and added a note to the paper that we were aiming for sample budgets that correspond roughly to similar runtimes (with the exception of PPO 200m, which we present to highlight that training PPO for longer does not result in competitive performance.)
> >
> > Note that we do not believe that REPPO will always outperform SAC, as this depends on many complex factors, and in general, it will have significantly worse sample efficiency, as is common for on-policy algorithms. We removed the claim from the manuscript and added additional context in the wall-clock time section.
> >
> > [1] Suh et al. "Do Differentiable Simulators Give Better Policy Gradients?", ICML 2022, https://proceedings.mlr.press/v162/suh22b/suh22b.pdf

---

> > ### Comment · Reviewer_XoJ7 · 2025-11-21
> > **Reply to authors' response**
> >
> > 1. If the authors agree with my thought that the brought benefit in Figure 3 was more because of the on-policy actions rather than difference between a pathwise vs. score-based estimator in this case, I think the way the current paper is framed almost exclusively claims that the main benefit of their approach comes from the ability of using a pathwise estimator, which is not the true benefit source (figure 3 directly proves against the main story where a score-based estimator is only marginally worse, whereas both REPPO (pathwise and score-based) are significantly better than estimators that involve updates on the off-policy actions).
> >     - Just to be clear, I do think the findings in this paper are valuable and interesting, I am just saying a deeper investigation might be due to understand what brings the benefit
> >
> > 2. I think the bias-variance trade-off is exactly the most important part that dues deeper discussion in this work:
> >     - First, compared with PPO, the cost of REPPO is not that “REPPO now uses a biased approximation of the policy gradient due to a surrogate Q function, whereas PPO does not”. PPO itself already introduces bias to REINFORCE type of estimators, e.g., via importance weight clipping, using GAE for return estimation.
> >     - Therefore, the main question is “which kind of bias and when brings more benefit”— We understand sufficiently how GAE trades off between bias and variance via a bootstrapping horizon. However, in this work, only positive empirical evidence is shown that REPPO **can** work better than PPO, i.e., using a learned Q to estimate a policy gradient can be better than using a GAE return estimator. However, a more valuable (theoretical and/or empirical) analysis is important to understand when this conclusion **breaks** (and I personally think this should not be reserved for future work).
> >
> > 3. Other than the points above, I agree that the paper is interesting and sheds light on an under-explored, potentially important direction. Thanks for the effort in addressing my comment. Just because of the points above, though, I will keep my original evaluation for now.

---

> ### Author Response · Authors · 2025-11-21
> **Question**
>
> Dear reviewer, thank you for your reply. As a very quick comment/question:
>
> We are genuinely not sure how we could show/prove when which kind of bias is better/more acceptable in the general context of policy gradient estimation. As we are dealing with the approximation ability of neural networks, making any concrete theoretical claims is likely impossible given the state of the field.
> As for empirical claims, this again seems to depend a lot on the exact domain.
>
> In the interest of strengthening the paper (or follow up work, independent on the status of this conference submission) we are genuinely interested in suggestions or ideas on what you would like to see! What could an experiment look like that answers your question satisfactory? We have so far not found a scenario in which REPPO strongly underperforms PPO (aside from a few individual DMC environments on which our single hp set is slightly suboptimal).
>
> We are not trying to bargain here, we are really just interested if you have concrete ideas.

---

### Official Review · Reviewer_uFnn · 2025-10-27

**Soundness:** 3
**Presentation:** 3
**Contribution:** 3
**Rating:** 4
**Confidence:** 4

**Summary:**

This paper proposes the REPPO algorithm that optimizes the policy by directly differentiating the Q function in a completely on-policy setting.

**Strengths:**

+ The problem studied in this paper is motivated well.

+ Presentation is clear and easy to follow.

+ Empirical results are comprehensive and the proposed algorithm performs well across several benchmark environments. It is particularly interesting to see that directly optimizing the $Q$ function is feasible for on-policy methods, which is often believed to rely on off-policy data.

**Weaknesses:**

I have some questions regarding the design of the REPPO algorithm.

+ Upon checking the code, I found that the critic function does not directly take the state-action pair as the input, but a latent state $\phi(s, a)$ which is the output of a feature module. That is, when differentiating the policy objective (Equation 12), using the reparameterization trick it yields
$$ \nabla_\theta Q(x_i, a) = \nabla_\theta Q_\eta ( \phi(s_i, a) ) = \frac{\partial Q}{\partial \eta} \frac{\partial \phi}{\partial a} \frac{\partial a}{\partial \theta}$$
where I denote the critic parameters by $\eta$ instead of $\phi$ to avoid abusing notation. I wonder if the feature module $\phi$ is critical to the algorithm as the value landscape is much simpler in the latent space.

+ Can you provide more insights on why the HL-Gauss loss is better than MSE in critic training?

+ The rollout length in the code is 128, which is longer than horizons commonly used in on-policy methods. I wonder how different lengths affect the performance, especially for the tasks with short horizon such as Maniskill manipulation problems.

+ Training the Q function can be numerically unstable, which has motivated many tricks in off-policy methods such as double Q learning, employing target networks, etc.. In Section 3.3, the paper introduces three methods to enhance the stability. However, as shown in the ablation study (Figure 10), removing any of them does not lead to da significant performance drop. Can you give me some intuition on why the critic training in REPPO is more robust than its off-policy counterparts beyond these three points?

+ Typically, directly differentiating the Q function requires it to have a nice first-order approximation to the local value landscape in the state-action space so that its gradient with respect to the action yields a correct updating direction. To achieve this, off-policy methods use very short horizons for the TD estimation, such as one step in TD3 and SAC and five steps in TDMPC, under the consideration that longer trajectories would introduce more noise into the estimation and affect the accuracy of the gradient estimation. As mentioned earlier, REPPO uses a very long TD horizon, and I wonder why its Q function can still effectively capture the gradient direction in this case.

I would be happy to increase my score if the authors can address my concerns.

**Questions:**

Please see Weaknesses.

---

> ### Author Response · Authors · 2025-11-17
> **Review reply**
>
> Dear reviewer,
>
> we are happy to answer the questions you raise.
> If there are any further improvements we can make, we are also happy to revisit and update the draft.
>
> To answer your questions in detail:
>
> > Upon checking the code, I found that the critic function does not directly take the state-action pair as the input, but a latent state $\phi(s, a)$ which is the output of a feature module.
>
> On a code level, we concatenate the observation and action vectors and use two neural network modules. This modular structure allows us to extract intermediate representations for the auxiliary loss (see Section 3.3 and Appendix D.3).
> However, without the auxiliary loss, the architecture is identical to a standard neural network, except that it is split into two modules in the code, which does not change the actual computation.
> The gradient is propagated without any further adjustments through both modules.
> Therefore, this design is not critical to the performance of REPPO.
>
> > Can you provide more insights on why the HL-Gauss loss is better than MSE in critic training?
>
> Prior work [1] has shown that framing regression as classification and optimizing a categorical cross-entropy loss can significantly improve the performance and scalability of value-based RL methods.
> Other, more recent papers have similarly found improved performance using various categorical losses [3,4,5,6]. Of these, [3] offers an explanation and we have added a citation to the manuscript. Note that [3] had not been publicized when we submitted the original manuscript.
>
> > The rollout length in the code is 128, which is longer than horizons commonly used in on-policy methods. I wonder how different lengths affect the performance, especially for the tasks with short horizon such as Maniskill manipulation problems.
>
> In our experiments, we generally observe that longer rollout horizons improve performance, primarily because they provide more accurate value targets. Similar observations were reported by Andrychowicz et al. [2]. We include ablations with a shorter horizon of 32 steps in the Appendix (Figure 10b).
>
> We also note that rollout horizons vary substantially across the community. For example:
> - PPO in SB3 and CleanRL uses a default horizon of 2048 (albeit without large-scale paralellization of environments).
> - In mujoco_playground, rollout lengths range from 10 to 100.
> - In ManiSkill, PPO baselines are run with horizons ranging from 4 to 100 (compare https://github.com/haosulab/ManiSkill/tree/main/examples/baselines/ppo).
>
> > Training the Q function can be numerically unstable, which has motivated many tricks in off-policy methods such as double Q learning, employing target networks, etc. ...
>
> As shown in Figure 10, much of REPPO’s performance and stability arises from the interplay of several design choices, including the adaptive KL regularization and the categorical critic architecture. Most importantly, REPPO benefits from being a fully on-policy formulation, which naturally avoids several sources of instability common in off-policy algorithms, i.e. the effects of the “deadly triad.” In this sense, our method is more closely related to SARSA than classical Q learning.
>
> > Typically, directly differentiating the Q function requires it to have a nice first-order approximation to the local value landscape in the state-action space so that its gradient with respect to the action yields a correct updating direction.
>
> Note that we do not differentiate an n-step rollout like done in TD-MPC or SVG, but use the same gradient as TD3 or SAC w.r.t to the action. The multi-step targets are used as regression targets for the Q function, and simply follow the intuition of TD-$\lambda$, trading off bias and variance between a bootstrapped and a Monte-Carlo target.
> Further, as we are on-policy, we do not have to account for complications like V-trace and other importance sampling corrections, which often drastically increase the variance.
> We are unaware of any research that shows that the step length of the TD target substantially changes the gradient landscape, but we are happy to discuss references if the reviewer knows of any.
>
> [1] Farebrother et al.  (2024). Stop Regressing: Training Value Functions via Classification for Scalable Deep RL. <i>International Conference on Machine Learning</i>
>
> [2] Andrychowicz, M. et al.  (2021). What matters for on-policy deep actor-critic methods? a large-scale study. *International Conference on Learning Representations*
>
> [3] Palenicek et al. (2025), XQC: Well-conditioned optimization accelerates deep reinforcement learning, arXiv
>
> [4] Nauman et al. (2025), Bigger, Regularized, Categorical: High-Capacity Value Functions are Efficient Multi-Task Learners, NeurIPS
>
> [5] Hansen et al. (2024), TD-MPC2: Scalable, Robust World Models for Continuous Control, ICLR
>
> [6] Hafner et al. (2025), Mastering diverse control tasks through world models, Nature

---

> > ### Comment · Reviewer_uFnn · 2025-11-21
> > **Reply to Rebuttal**
> >
> > Thanks for the detailed response, now I have a better sense of what your algorithm does. As mentioned by Reviewer XoJ7, it should be made more precise how REPPO actually estimates values. Overall it is a solid paper with interesting ideas and I have raised my score, good luck.

---

> > > ### Author Response · Authors · 2025-11-25
> > > **How REPPO actually estimates values**
> > >
> > > Dear reviewer, thank you for your comment. A quick request for clarification: How could we make it more clear "how REPPO actually estimates values"? We are somewhat unsure we understand the question correctly, and would really like to address any weaknesses in the paper.
> > >
> > > REPPO uses an on-policy TD-lambda scheme, similar to most on-policy algorithms that we are aware of. The only major departure from prior work is that we estimate state-action value functions (Q) instead of state value functions (V). We also added some architectural advances, but while those are relevant for training with a pathwise or Q-based estimator, they do not significantly boost the performance of the ratio based estimator used in PPO (compare Figure 3).

---

### Official Review · Reviewer_nRqE · 2025-10-29

**Soundness:** 3
**Presentation:** 2
**Contribution:** 3
**Rating:** 6
**Confidence:** 4

**Summary:**

The authors are motivated by the goal of formulating a modern on-policy reinforcement learning algorithm that dispenses with the large replay buffers characteristic of many state-of-the-art off-policy methods, thereby reducing the associated memory footprint and data management overhead. To this end, they propose Relative Entropy Pathwise Policy Optimization (REPPO), an on-policy actor-critic algorithm that builds upon the maximum entropy framework and utilizes the pathwise policy gradient, similar to Soft Actor-Critic (SAC).

The core of their contribution lies in adapting these off-policy components to the more challenging (wrt value-function learning) on-policy setting. In particular, they replace the standard single-step TD targets with multi-step TD(λ) value targets. This is a critical choice, as it aims to reduce the bias in value estimates that is particularly severe in the on-policy regime due to the high temporal correlation of the training data. Furthermore, to ensure stable policy updates, the authors employ a KL-regularized trust region, dynamically tuning the Lagrange multipliers using a simultaneous gradient-based approach inspired by SAC. Finally, the algorithm incorporates several recent advances in value function learning to further enhance stability, including the use of a categorical Q-function representation, layer normalization, and auxiliary tasks. The authors provide a thorough empirical evaluation comparing REPPO against several state-of-the-art algorithms, reporting significant performance improvements in terms of final performance and stability.

**Strengths:**

*   The paper presents a compelling and timely approach to on-policy learning, which is of significant interest to the community, particularly in the context of modern, massively parallelized simulation environments where data generation is no longer the primary bottleneck.
*   The proposed method is a well-motivated synthesis of several powerful, existing algorithmic components. The originality lies in their successful adaptation and integration for the on-policy setting.
*   The claims are supported by an extensive empirical evaluation, including thorough ablation studies, which demonstrates strong performance against relevant baselines.

**Weaknesses:**

The primary weaknesses of the paper lie in its exposition, where the mathematical notation could be more precise and consistent to improve clarity.
*   The notation in Equation (1) is slightly ambiguous. The expectation is written as being over the state distribution $s \sim \mu_\pi$, but the term inside also depends on the action $a$. It should be made explicit that the action is sampled from the policy, i.e., $a \sim \pi_\theta(\cdot|s)$.
*   The presentation of the pathwise policy gradient in Equation (2) could be clearer:
    *   The notation $a = \pi_\theta(x)$ is used without introduction. This notation typically implies a deterministic policy, whereas the algorithm employs a reparameterized stochastic policy. This should be clarified to avoid confusion with the Deterministic Policy Gradient theorem.
    *   Furthermore, the expectation over the noise variable $\epsilon$, which is essential to the reparameterization trick, is not made explicit in the expression.
*   There is a notational inconsistency between different parts of the paper. For instance, Equations (8-11) use $\pi_\theta(x)$ to denote the policy distribution, while Equation (12) uses the more explicit conditional notation $\pi_\theta(a|x)$. Consistent notation should be used throughout.

Typos/Nitpicks:
*   In the caption for Figure 4, it is stated that "we report the final performance at 100 million steps." However, the corresponding plot in Figure 4b includes a curve for "PPO (200M)". Please clarify the exact training horizon being reported.
*   Line 368: "perforamnce" should be "performance".

**Questions:**

*   A minor but interesting point from Figure 2 is that the pathwise gradient with the ground truth objective appears to converge more slowly than with the strong surrogate. Could the authors elaborate on the potential reasons for this phenomenon?
*   Regarding the policy update, the authors supplement the automatic multiplier tuning with a heuristic clipping of the actor loss (Equation 12). It would be valuable for the authors to comment on the empirical importance of this clipping mechanism. How does it contribute to the algorithm's overall stability and performance?
*  I'd suggest improving on the points mentioned under "Weaknesses".

---

> ### Author Response · Authors · 2025-11-17
> **Review reply**
>
> Dear reviewer,
>
> we are happy to clarify the notation of our paper following your suggestions, and we hope this improves the draft.
> We are happy to further revise the draft in case any further suggestions for improvement are raised.
>
> > The primary weaknesses of the paper lie in its exposition, where the mathematical notation could be more precise and consistent to improve clarity.
>
> We are happy to update the notation as suggested. Please refer to our rebuttal revision and let us know if there are any other notational inconsistencies that we missed.
>
> > The notation in Equation (1) is slightly ambiguous. The expectation is written as being over the state distribution $s \sim \mu_\pi$, but the term inside also depends on the action . It should be made explicit that the action is sampled from the policy, i.e., $a \sim \pi_\theta(\cdot|s)$.
>
> > The presentation of the pathwise policy gradient in Equation (2) could be clearer:
> > - The notation $a = \pi_\theta(x)$ is used without introduction. This notation typically implies a deterministic policy, whereas the algorithm employs a reparameterized stochastic policy. This should be clarified to avoid confusion with the Deterministic Policy Gradient theorem.
> > - Furthermore, the expectation over the noise variable $\epsilon$, which is essential to the reparameterization trick, is not made explicit in the expression.
>
> Thanks for pointing this out. We have now provided the equations for both the original deterministic policy gradient (as readers might be more familiar with it), and the reparameterized pathwise policy gradient.
> We have also clarified where we speak about a deterministic, stochastic, or reparameterized policy.
>
> > There is a notational inconsistency between different parts of the paper. For instance, Equations (8-11) use $\pi_\theta(x)$ to denote the policy distribution, while Equation (12) uses the more explicit conditional notation $\pi_\theta(a|x)$. Consistent notation should be used throughout.
>
> > Typos/Nitpicks:
> > - In the caption for Figure 4, it is stated that "we report the final performance at 100 million steps." However, the corresponding plot in Figure 4b includes a curve for "PPO (200M)". Please clarify the exact training horizon being reported.
>
> This is indeed a typo in the plot, for maniskill the original authors (Stone et al.) provide results up to 100m steps, which we use (compare https://wandb.ai/stonet2000/ManiSkill/reports/PPO-Results--VmlldzoxMDQzNDMzOA).
>
>
> ## Questions
> We now answer your additional questions in detail.
>
> > A minor but interesting point from Figure 2 is that the pathwise gradient with the ground truth objective appears to converge more slowly than with the strong surrogate. Could the authors elaborate on the potential reasons for this phenomenon?
>
> This is indeed a curiosity. We did not expect this phenomenon to occur and debated changing the figure, as it is likely function and starting point dependent. However this was the first setup we tried and so we did not feel it would be a good idea to keep searching configurations until the figure looked the way we expected it to. Similar phenomena can be found in the literature on comparing surrogate approximations to differentiable simulators [1]. Surrogate functions can smooth out the original, which can lead to slightly accelerated convergence of gradient based approaches. However, we do not believe that surrogate functions will lead to faster convergence in general.
>
> If the reviewers believe that this figure is too confusing or potentially misleading we are happy to update it.
>
>
> > Regarding the policy update, the authors supplement the automatic multiplier tuning with a heuristic clipping of the actor loss (Equation 12). It would be valuable for the authors to comment on the empirical importance of this clipping mechanism. How does it contribute to the algorithm's overall stability and performance?
>
> We are happy to add an additional ablation to the appendix. This will take some additional time and we hope to have the results for you by the end of the week.
>
> [1] Suh et al. "Do Differentiable Simulators Give Better Policy Gradients?", ICML 2022, https://proceedings.mlr.press/v162/suh22b/suh22b.pdf

---

### Official Review · Reviewer_EJRD · 2025-10-31

**Soundness:** 3
**Presentation:** 4
**Contribution:** 3
**Rating:** 8
**Confidence:** 4

**Summary:**

The paper proposed utilising the pathwise gradient estimator with an accurately learned surrogate value function to construct more efficient on-policy algorithms. The approach is empirically evaluated across multiple datasets to demonstrate its effectiveness.

**Strengths:**

- The paper is well-written and easy to follow.
- Figure 2 shows Illustrative examples of different estimators (policy gradient estimator vs. Different pathwise gradient estimators).
- Experiments contain various benchmarks (DMC suite and ManiSkill environment) and baselines.
- The efficiency of the algorithm is shown in different settings: number of samples to reach a certain performance level, memory, and wall-clock time.

**Weaknesses:**

See the questions section.

**Questions:**

Minors: the meaning of $x’_i$ in equation (6)?

---

> ### Author Response · Authors · 2025-11-17
> **Review reply**
>
> Dear reviewer,
>
> we are happy to see that you appreciate our paper.
> We have added a new revision to answer some of the suggestions made by you and the other reviewers.
> We are happy to further revise the draft in case any further suggestions for improvement are raised.
>
> > Minors: the meaning of $x'$ in equation (6)?
>
> The empirical next state $x'\sim p(\cdot|x,a)$ obtained from the rollout of the policy in the environment. We clarified the notation in the paper.

---

### Author Response · Authors · 2025-11-17
**Rebuttal draft updated**

Dear reviewers,

We are happy to see the overall positive reception of our paper, and we greatly appreciate your suggestions for improvements.
We have uploaded a revised draft of the paper, taking care to address your comments.
The largest change is an additional pedagogical example for Figure 2, which shows the impact of importance sampling on gradient variance.
As reviewer XoJ7 correctly pointed out, removing importance sampling is a major factor in the strong performance of REPPO, so we wanted to highlight it more strongly in the introduction.

We have also further streamlined and clarified the notation, following reviewer nRqE's suggestions and comments.

Finally, we are currently working on two additional ablations/comparisons concerning the KL and the clipping heuristic introduced in Eq. 13.
We will share these by the end of the week.

To make the changes clear and easy to survey, we have uploaded a diff of the changes in the PDF.

We hope we have addressed all points in the reviews and are happy to engage in further discussion if any unclear points remain.

Kind regards, the authors

---

> ### Comment · Area_Chair_D3Nn · 2025-11-21
> **Author-Reviewer Discussion**
>
> Dear reviewers,
>
> Please review the authors' response and adjust your rating accordingly. If you have any further questions, please discuss with the authors further.
>
> AC

---

### Public Comment · ~Zhengpeng_Xie1 · 2025-11-25

Great work. I noticed that you cited SPO [1]. We also found that PPO struggles to perform well in environments like **Humanoid**. Would it be possible for you to include SPO as an additional baseline in Figure 14?

[1] Z Xie et al. Simple policy optimization. ICML 2025.

---

### Author Response · Authors · 2025-11-25
**Additional ablation**

Dear reviewers,

As promised, we have expanded the ablation in Figure 10 a) to include the KL clipping heuristic and the reverse KL.

Kind regards, the authors

---

### Author Response · Authors · 2025-11-29
**Final rebuttal comment**

Dear reviewers,

Thank you for your help in improving our submission draft. Even though there is no more opportunity for you to provide feedback, we wanted to make on last comment detailing all changes during the rebuttal process, and highlighting the changes we made to our draft. We appreciated the feedback and your help.

Dear new AC, we will briefly summarize what improvement suggestions were made and how we implemented them.

As reviewer nRqE requested we have unified and straightened out the notation of the paper.

For reviewer uFnn, we clarified the role of the encoder in the code, and why our algorithm achieves better stability than off-policy Q learning. We also added a new experiment ablating the direction of the KL and the piecewise clipping variant of the REPPO loss function.

There was a longer discussion with reviewer XoJ7 about two core points: 1) the relative importance of pathwise policy gradients vs forgoing importance sampling and 2) understanding the relative strengths and weaknesses of REPPO wrt to PPO better.

To address 1), we made some careful yet crucial changes to the writing throughout the paper to contrast not just "pathwise vs score-based PG" but "using a trained Q function vs PPO style AC learning". We agree that this shift makes our contributions and the takeaways for the community much more clear.

To address 2) we conducted a new experiment varying the size of the data batch collected in each cycle of the algorithm. This is the new section in the appendix. We see a clear difference between REPPO and PPO: In REPPO, a larger data batch is generally better, while PPO struggles to fully make use of a large buffer. We conjecture that there are two important effects: a larger batch allows the Q function trained for REPPO to generalize better, while a smaller batch means the algorithm has to do less update steps on stale data before new data is obtained. REPPO and PPO behave differently in how sensitive they are to these choices. Note that all algorithms see the same number of data samples in total, and do the same number of gradient steps with the same minibatch size. Only the frequency of gathering new on-policy data is varied.
Note that both this writing update and the new experiment were made today, after the reviewers final comment, so the reviewer did not have a chance to reply to it.

We acknowledge that the reviewers cannot comment anymore on whether this new experiment addresses the remaining points discussed about understanding the performance gains.
We nonetheless believe our further experiments will provide a good avenue to further understand the characteristics of different on-policy algorithms by further analyzing these critical parameters.

We are happy to clarify any further questions from the AC in the final days of the rebuttal process. Note that we currently uploaded two concatenated versions of the paper. The first is a diff to highlight changes made since the beginning of the rebuttal period, the second is the updated draft to show that we are within page limit.

Kind regards, the authors

---

### Meta-Review · Area_Chair_Fo7P · 2026-01-06

**Summary:**

This paper proposes Relative Entropy Pathwise Policy Optimization (REPPO), an on-policy actor-critic algorithm that builds upon the maximum entropy framework and utilizes the pathwise policy gradient to eliminate the large replay buffers and data-management overhead typical of state-of-the-art off-policy methods.

The main contributions are: (1) This paper adapts these off-policy components to the more challenging (in terms of value function learning) on-policy setting. In particular, the authors replace the standard single-step TD targets with multi-step TD(λ) targets to reduce the bias in value estimates that is particularly severe in the on-policy regime. (2) Moreover, to ensure stable policy updates, the authors employ a KL-regularized trust region, dynamically tuning the Lagrange multipliers using a simultaneous gradient-based approach inspired by SAC.
(3) The authors also incorporate several enhancements for value function learning to further enhance training stability, such as categorical Q-function representation, layer normalization, and auxiliary tasks. A thorough empirical evaluation comparing REPPO against several state-of-the-art algorithms is provided to show improvements in terms of final performance and stability.

The main strengths of this work:
- The problem studied in this paper is well-motivated. This work showcases that using a pathwise policy gradient with a properly learned state-action value function can be effective in on-policy RL, obviating the need for a replay buffer or importance sampling.
- It is interesting to see that directly optimizing the Q function is feasible for on-policy methods, which is usually believed to require off-policy data, as mentioned by multiple reviewers.
- The claims are well supported by comprehensive empirical evaluation, including extensive ablation studies, which demonstrates strong performance against the baselines.

On the other hand, the reviewers also raised a few concerns and key questions:
- Some algorithmic design and experimental results need more justification and explanation: (1) In the ablation study (Figure 10), removing any of them does not lead to significant performance drop. Some explanation on why the critic training in REPPO is more robust than its off-policy counterparts is needed. (2) Regarding the policy update, REPPO supplements the automatic multiplier tuning with a heuristic clipping of the actor loss (Equation 12). (3) Explain the TD horizon used in REPPO for the direct differentiation of the Q function.

- More in-depth explanation on where the main performance gain of REPPO comes from is needed as the performance difference between REPPO (pathwise) and REPPO (score-based, Q) is actually tiny in Figure 3. The reviewer also specifically asked about whether the gain comes from the pathwise gradient or the on-policy actions.

- Clarity and exposition: The reviewers noted that the mathematical notations are quite inconsistent and requested for clearer definitions and more consistent notation to improve readability.

**Reviewer Concerns:**

After taking a careful look at the reviews and the rebuttal response, I found most of the concerns appear either fully addressed or alleviated.

One remaining point is on whether the performance gain of REPPO comes mainly from the pathwise gradient or the on-policy action resampling (the removal of importance sampling). In the discussion phase, the authors and the reviewer have reached a consensus that both can be contributing factors to the overall gain, and the authors have updated the paper accordingly to reflect this.

**Reviewer Scores:**

In the initial reviews, the reviewers' evaluations are mostly positive (EJRD: 8 / nRqE: 6 / uFnn: 4 / XoJ7: 6).

After the rebuttal, most of the concerns have been addressed, and all the reviewers agree unanimously that this paper is interesting and a valuable contribution that sheds light on an underexplored direction in on-policy RL.

---

### Decision · Program_Chairs · 2026-01-26

Accept (Poster)